# Development of pathophysiologically relevant models of sickle cell disease and β-thalassemia for therapeutic studies

Pragya Gupta[1,2,6], Sangam Giri Goswami[1,2,6], Geeta Kumari[3,6],
Vinodh Saravanakumar[1], Nupur Bhargava[1], Akhila Balakrishna Rai ®[4],
Praveen Singh[1,2], Rahul C. Bhoyar[1], V. R. Arvinden[1,2], Padma Gunda[5], Suman Jain[5],
Vanya Kadla Narayana ®[4], Sayali C. Deolankar ®[4], T. S. Keshava Prasad ®[4],
Vivek T. Natarajan ®[1,2], Vinod Scaria[1,2], Shailja Singh ®[3] ✉ &
Sivaprakash Ramalingam ®[1,2] ✉

Ex vivo cellular system that accurately replicates sickle cell disease and β-thalassemia characteristics is a highly sought-after goal in the field of erythroid biology. In this study, we present the generation of erythroid progenitor lines with sickle cell disease and β-thalassemia mutation using CRISPR/Cas9. The disease cellular models exhibit similar differentiation profiles, globin expression and proteome dynamics as patient-derived hematopoietic stem/progenitor cells. Additionally, these cellular models recapitulate pathological conditions associated with both the diseases. Hydroxyurea and pomalidomide treatment enhanced fetal hemoglobin levels. Notably, we introduce a therapeutic strategy for the above diseases by recapitulating the HPFH3 genotype, which reactivates fetal hemoglobin levels and rescues the disease phenotypes, thus making these lines a valuable platform for studying and developing new therapeutic strategies. Altogether, we demonstrate our disease cellular systems are physiologically relevant and could prove to be indispensable tools for disease modeling, drug screenings and cell and gene therapy-based applications.

β-hemoglobin disorders, such as sickle cell disease (SCD) and β-thalassemia (BT), are the most common inherited monogenic blood disorders globally. They impose a significant burden on patients, their families, and the healthcare system due to high morbidity and mortality rates. Despite decades of research, there are only four FDA-approved medications available for the management of SCD with hydroxyurea (HU) being the most widely used drug that partially benefits patients by inducing fetal hemoglobin (HbF) production.

However, HU has potential side effects and requires additional interventions for disease management[1]. In contrast, Luspatercept is the only approved drug currently available for β-thalassemia patients. At present, the most curative approach available worldwide for these disorders is allogeneic hematopoietic stem cell transplantation (HSCT), which is limited by donor availability and comes with side effects such as graft versus host disease (GvHD). Recently, the FDA approved Zyntelgo as the first cell and gene therapy-based product for

[1]CSIR- Institute for Genomics and Integrative Biology, Mathura Road, Sukhdev Vihar, New Delhi, India. [2]Academy of Scientific and Innovative Research (AcSIR), Ghaziabad 201002, India. [3]Special Center for Molecular Medicine, Jawaharlal Nehru University, New Delhi, India. [4]Center for Systems Biology and Molecular Medicine, Yenepoya Research Centre, Yenepoya (Deemed to Be University), Mangalore 575018, India. [5]Thalassemia and Sickle Cell Society- Kamala Hospital and Research Centre, Shivarampally, Hyderabad, India. [6]These authors contributed equally: Pragya Gupta, Sangam Giri Goswami, and Geeta Kumari.
✉e-mail: shailjasingh@mail.jnu.ac.in; sivaramalingam@igib.res.in

treating BT. Notably, Casgevyy, a pioneering genome-editing-based cell therapy product, received approval very recently for the treatment of both SCD and BT[2,3]. Additionally, Lyfgenia, a lentiviral-based cell therapy product, is approved for treating patients with SCD[4].

Preclinical in vivo studies have become crucial in understanding erythropoiesis and studying red blood cell diseases[5]. However, it is becoming increasingly clear that fundamental differences exist between human and mouse erythropoiesis, highlighting the limitations of in vivo models. To overcome these limitations, erythroid cells produced through in vitro erythropoiesis by differentiating primary hematopoietic stem and progenitor cells (HSPCs) are frequently employed. A major bottleneck in this approach is that HSPCs have a restricted capacity for self-renewal and are further limited by access to human donors[6]. Hence, collecting erythroid progenitor cells from these patients for studying disease mechanisms and drug screening is not feasible. Given the scarcity of human primary cells from healthy and diseased donors, generating such valuable erythroid progenitors for large-scale applications has remained challenging. Though several studies have generated multiple iPSC from SCD and BT patients, erythroid cells derived from these patient-derived iPSCs mainly exhibit an embryonic/fetal globin expression pattern with inefficient or aberrant terminal maturation, rather than adult-stage hemoglobin expression, making them unsuitable as cellular model lines[7–9].

Despite the promise of emerging cell and gene therapies, translation and access to large populations of patients residing in low-income countries such as Sub-Saharan Africa and Southeast Asia, where β-hemoglobinopathies are prevalent and basic medical treatment remain elusive[10]. As a result, continuous efforts are required for development of alternative treatment strategies specially drug discovery and drug repurposing for clinical induction of γ-globin, anti-sickling and for rescuing globin chain imbalance. However, one factor that is a major hurdle in drug discovery for β-hemoglobinopathies is lack of a reliable and physiologically relevant cellular system for studies and screening of pharmacological small molecules for either SCD or BT.

Recent studies have demonstrated the establishment of immortalized erythroid progenitor cells such as Human Umbilical cord Derived Erythroid progenitor −2 (HUDEP-2) from cord blood, Bristol Erythroid Line -Adult (BEL-A) from adult[11] CD34+ cells and peripheral Blood immortalized erythroid progenitor cells PBiEPCs[12] from peripheral blood using lentiviral-based tetracycline-inducible HPV16-E6/E7 construct. Of these lines, the BEL-A line has exhibited significantly higher reticulocyte yield, enucleation rates, and more accurately reproduces adult erythropoiesis[11]. Since these lines were all derived from healthy donor cells, they don't recapitulate the disease phenotypes.

To our knowledge, there are currently no well-characterized erythroid progenitor cell lines available that can accurately replicate the pathophysiology of SCD and BT while also having the same genetic background apart from the disease mutation enabling consistency and reproducibility. To address this issue, we used a CRISPR/Cas9 coupled with piggyBac-based footprint-free approach, to generate immortalized erythroid progenitor cells, named BEL-A SCD mutation (BEL-A SCM) with a sickle cell mutation (Codon 6, A > T) and BEL-A BT mutation (BEL-A BTM) with prevalent β-thalassemia mutation (IVS1-5, G > C), respectively. Differentiated erythroid cells from these cell lines exhibited similar differentiation profile, globin expression and proteome dynamics to those derived from patient's HSPCs. We have shown that reticulocytes derived from BEL-A SCM exhibit resistance to *P. falciparum* invasion comparable to SCD patient-derived RBCs. In this study, we additionally present the demonstration of the CRISPR-mediated generation of a beneficial HPFH3 genotype[13] in the β-globin cluster in both disease lines with therapeutically relevant levels of HbF that were sufficient to reverse the phenotypes of BT and SCD. It provides a vital preclinical data supporting a promising approach for the

treatment of β-hemoglobin disorders. Our physiologically relevant cellular systems provide a plethora of avenues for researchers to investigate various applications related to parasite invasion, drug validation, and genome-editing in the context of SCD and BT.

## Results

### CRISPR/Cas9-coupled piggyBac-mediated footprint-free strategy generates erythroid progenitor lines for SCD and BT

The experimental strategy used for generation, functional characterization, and applications of BEL-A SCD and BEL-A BTM is outlined in Fig. 1A. In this study, we used CRISPR/Cas9-coupled with piggyBac transposon system to introduce the biallelic SCD and BT mutations into BEL-A cells[14]. The strategy for introducing the SCD and BT mutations in BEL-A cells is shown in Supplementary Figs. 1, 2. In the first step, we utilized a dual DONOR strategy to target the conversion of wild-type alleles to the mutant alleles and generate homozygous SCD and BT (IVS1-5; G > C) lines. The clonal eGFP+ and dTomato+ cells were screened using flow cytometry and integration at the correct locus in three clones was confirmed through junction PCR and Sanger sequencing (Supplementary Figs. 3, 4). In the second step, we obtained homozygous BEL-A SCM and BEL-A BTM lines by piggyBac transposon-mediated excision of PSM. Negative selection for eGFP and dTomato was carried out by FACS and the genotype of the resulting clones was confirmed through restriction digestion and further by Sanger sequencing (Fig. 1B, C, Supplementary Figs. 3, 4).

The disease lines were visually indistinguishable from BEL-A WT cells when observed under brightfield microscopy (Supplementary Fig. 5A). Additionally, Giemsa staining analysis showed that BEL-A SCM and BEL-A BTM cells were similar to BEL-A WT cells, depicting proerythroblast to early basoerythroblast stages (Supplementary Fig. 5B). These findings indicate successful generation of immortalized erythroid progenitor cellular models for SCD and BT.

### BEL-A SCM and BEL-A BTM recapitulate disease-specific erythropoiesis

We investigated if they underwent erythropoiesis in a manner similar to patient-derived HSPCs. To achieve this, we induced differentiation of BEL-A (WT, SCM, BTM) and HSPCs (WT, SCM, BTM) into erythroblasts and characterized them morphologically at the end of the differentiation process (Supplementary Fig. 5C). Using Giemsa staining, we identified different stages of the erythropoiesis; basoerythroblast, polychromatic erythroblast, orthochromatic erythroblast, reticulocyte and erythrocyte (Fig. 1D, Supplementary Fig. 6). Based on data analysis we inferred that the differentiation potential of the BEL-A SCM and SCM HSPCs was similar to that of BEL-A WT and WT HSPCs, respectively. We observed ineffective erythropoiesis, which is one of the characteristics of BT, in the erythroblasts derived from BEL-A BTM and BTM HSPCs. The number of reticulocytes was reduced to 17% and 13% in BEL-A BTM cells and BTM HSPCs cells as compared to 26% and 24% in BEL-A WT and WT HSPCs, respectively (Fig. 1D, Supplementary Fig. 6). We also observed a decline in cell number in BEL-A BTM as compared to BEL-A WT and BEL-A SCM from Day 4 onwards, whereas in SCM and BTM HSPCs, the reduction was observed from Day 8 onwards (Fig. 1E). We further confirmed the expression of key surface markers, CD71 and CD235a using flow cytometry at the end of differentiation as shown in Supplementary Fig. 7

Enucleation progression was monitored at regular time intervals and it was observed that erythroblasts from BEL-A BTM and BTM HSPCs lagged behind from day 8 and day 18, respectively. However, no significant difference was observed between BEL-A WT and BEL-A SCM as well as between WT HSPCs and SCM HSPCs, indicating concurrent enucleation among the erythroblasts derived from BEL-A disease lines and primary disease HSPCs (Fig. 2A). The enucleation efficiency was compared at the end of differentiation and our data revealed erythroblasts from BEL-A SCM (54%) and SCM

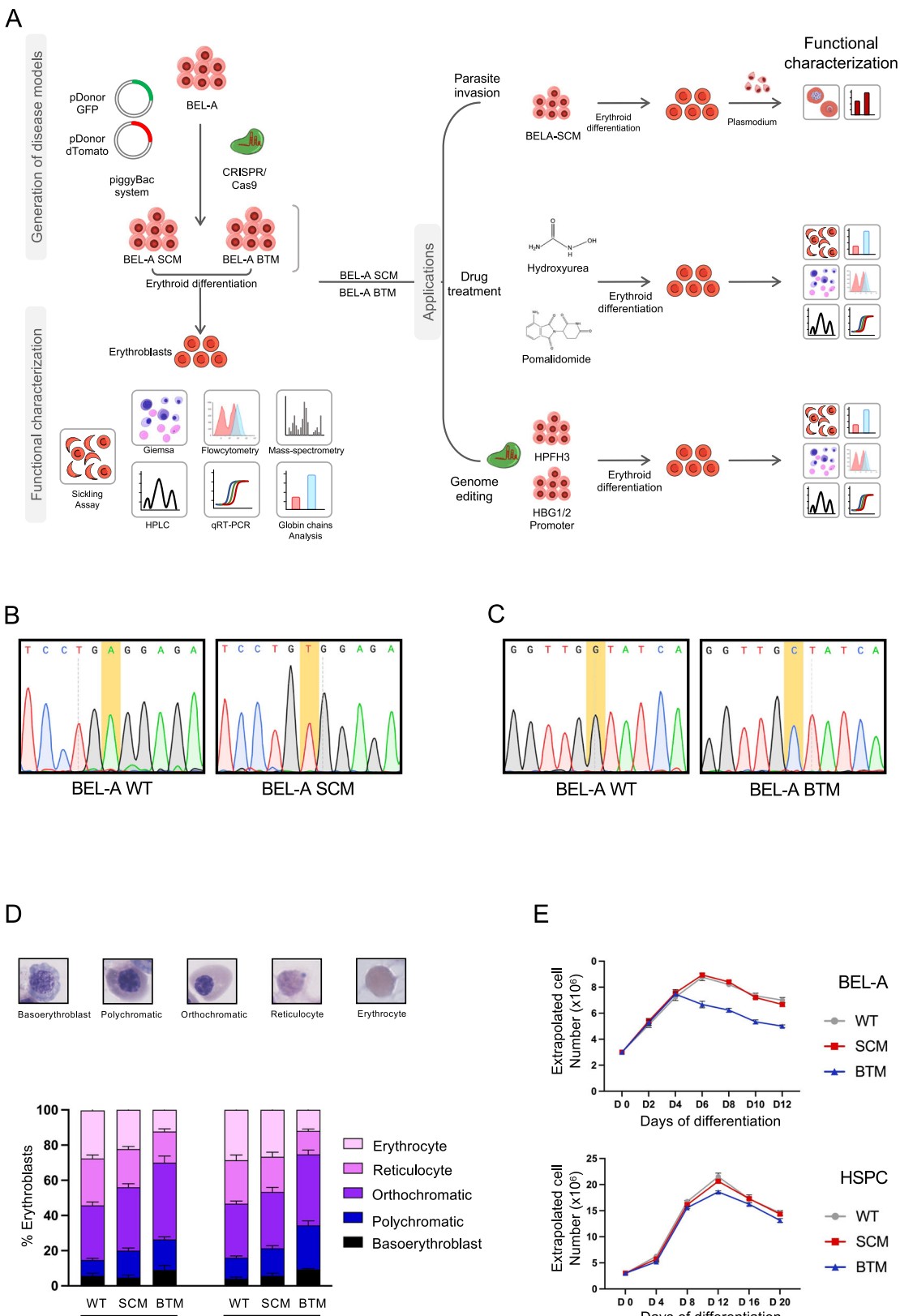

HSPCs (57%) showed almost equivalent enucleation efficiency as BEL-A WT (52%) and WT HSPCs (61%). In contrast, BEL-A BTM (31%) and BTM HSPCs (49%) derived erythroblasts showed a significant reduction in enucleation efficiency (Fig. 2B). These findings are in line with ineffective erythropoiesis, a well-established characteristic observed in BT.

## Evaluation of molecular characteristics in BEL-A erythroblasts derived from SCD and BT mutation

We aimed to investigate whether introducing disease mutations in BEL-A cells would result in higher basal levels of γ-globin as observed in patient cells. Relative γ-globin mRNA expression was found to be higher in BEL-A SCM (1.6-fold) and BEL-A BTM (2.3-fold) compared to

**Fig. 1 | Development of cellular disease model systems, BEL-A SCM and BEL-A BTM. A** Schematics showing the workflow for the development of disease model cell lines using CRISPR/Cas9 coupled with piggyBac Transposon system, functional characterizations, and their applications. **B** Chromatogram showing Sickle cell mutation (GAG > GTG) in BEL-A SCM through Sanger sequencing. **C** Chromatogram showing β-thalassemia mutation (IVS1-5, G > C) in BEL-A BTM through Sanger sequencing. **D** Representative Giemsa images from different erythroblasts present at the end of the differentiation in BEL-A cells (WT, SCM, BTM) and HSPCs (WT, SCM, BTM) respectively (≥200 cell counts per field). Graph is plotted as relative proportion of erythroblasts by total number of cells. **E** Cell expansion profile of BEL-A (WT, SCM, BTM) and HSPCs (WT, SCM, BTM) during differentiation. Cells were counted with trypan blue exclusion dye. All data is shown as Mean ± S.D from three independent replicates (n = 3). Source data is provided in the Source file.

WT but similar to the results in SCM (4.3-fold) and BTM HSPCs (2.9-fold) (Fig. 2C). We further checked the levels of β and δ-globin in BEL-A SCM and BEL-A BTM and observed a significant reduction of β and δ-globin expression (Fig. 2C). Therefore, we evaluated the intracellular HbF protein levels in erythroblasts and found higher levels of F-cells in BEL-A SCM (6%), BEL-A BTM (12%), SCM HSPCs (24%), and BTM HSPCs (22%) compared to BEL-A WT (1%) and WT HSPCs (7%) (Fig. 2D).

Our RP-HPLC analysis demonstrated that erythroblasts from BEL-A SCM and SCM HSPCs had 85% and 92% sickle Hb (HbS), respectively (Fig. 2E). RP-HPLC analysis showed globin chain imbalance in BEL-A BTM and BTM HSPCs (Fig. 2F), which is a well-established molecular characteristic of BT where there is a significant reduction or absence of β-globin chain resulting in globin chain imbalance.

To confirm that erythropoiesis in the disease-specific BEL-A lines reflect the disease phenotype and is not due to disruption of any key regulatory factors, we performed a qualitative analysis of key regulators using RT-PCR. The expression of *EKLF, GATA-1, LMO2, ZBTB7A, SOX-6, EKLF-3, FOG-1* and *BCL11A* in BEL-A SCM and BEL-A BTM were found to be comparable to disease-specific HSPCs (Supplementary Fig. 8). Overall, these results demonstrate that BEL-A SCM and BEL-A BTM lines recapitulate the phenotypes of respective primary SCM and BTM HSPCs.

## Comparative proteomic profiling of BEL-A SCM and BEL-A BTM with disease-specific HSPCs

Next, we wanted to investigate if our BEL-A SCM and BEL-A BTM demonstrate the molecular signatures associated with the disease. To eliminate any variation due to culturing conditions, we differentiated HSPCs in Phase I medium for 6 days and then transferred them to BEL-A expansion media without doxycycline for 2 days[15]. Flow cytometric analysis confirmed the stage-matching of BEL-A SCM and BEL-A BTM cells with their HSPCs counterparts (Supplementary Fig. 9). We performed Tandem mass tag (TMT) liquid chromatography mass spectrometry (LC-MS/MS) for a comparative analysis of BEL-A SCM and BEL-A BTM and its HSPC counterpart.

A total of 4943 proteins in BEL-A SCM versus SCM HSPCs and BEL-A BTM versus BTM HSPCs comparison were obtained through LC-MS/MS analysis of peptides derived from whole cell lysates (Supplementary Data 1). Proteomics data from earlier studies have also quantified around 5000 proteins at the proerythroblast stage[16]. The heatmap and PCA blot of BEL-A SCM versus SCM HSPCs (Fig. 3A, Supplementary Fig. 10A) and BEL-A BTM versus BTM HSPCs (Fig. 3B, Supplementary Fig. 10B) comparison group suggesting the overall pattern between both disease lines are similar to their HSPC counterparts. We considered fold change ≥1.5 for upregulation and fold change ≤0.67 for downregulation with *p*-value ≤ 0.05. Of 4943 proteins, only 45 proteins (<1%) and 18 proteins (<1%) were altered in BEL-A SCM versus HSPCs SCM (Fig. 3C) and BEL-A BTM versus HSPCs BTM (Fig. 3D) (Supplementary Data 2). As previously described, our proteome analysis also did not observe any protein expression specific to the BEL-A cells[11].

## BEL-A SCM and BEL-A BTM derived reticulocytes exhibit similar disease pathology as the erythroblasts derived from disease-specific HSPCs

After confirming the similarity in erythropoiesis and proteome profile between BEL-A SCM/BEL-A BTM and HSPC counterparts, we next investigated the manifestation of disease pathology in these cell lines. We exposed BEL-A SCM cells to hypoxia and found that 73% of the cells sickled, which closely paralleled 88% sickling observed in erythrocytes from SCD patients (Supplementary Fig. 11A, B). Similarly, we observed sickling in 44% of in vitro differentiated erythroblasts derived from SCM HSPCs (Fig. 3E, F) which correlates to the percentage of sickling and the percentage of enucleated cells (Fig. 2B). Next, we evaluated the ROS levels in our cellular disease model lines where BEL-A SCM (23%), BEL-A BTM (21%) and SCM HSPCs (21%), BTM HSPCs (29%) were found to have significantly higher ROS levels than BEL-A WT (3%) and WT HSPCs (5%) (Fig. 3G). These results were consistent with the ROS levels in respective blood samples (WT, SCD, BT) (Supplementary Fig. 11C).

To evaluate the presence of globin chain imbalance in BEL-A BTM, we measured the ratio of β-like globin and α-globin using flow cytometry and RP-HPLC. Flow cytometric analysis of the BEL-A BTM and BTM HSPCs-derived erythroblasts showed a significant reduction in β-like globin by 48% and 58% respectively (Fig. 3H). RP-HPLC data further showed a reduction of total β-like globin by 42% and 46% in BEL-A BTM and BTM HSPCs (Fig. 2F), which is a characteristic of BT. Overall, our results suggest that BEL-A SCM and BEL-A BTM exhibit similar disease pathology to disease HSPCs and are cellular model systems for SCD and BT.

## Erythroblasts differentiated from BEL-A SCM resist *Plasmodium* Invasion

To examine the effect of low oxygen concentration on invasion and growth of *P. falciparum* inside the BEL-A SCM reticulocytes, we first performed the invasion assay using BEL-A SCM-derived reticulocytes and compared it with SCD patients'-derived erythrocytes (SCD RBCs). Parasite invasion efficiency at 12 hpi (hours post-infection) demonstrated 20% and 33% reduction in BEL-A SCM and SCD RBCs, respectively (Fig. 4A, B). We then examined whether parasites that have invaded sickled RBCs have any effect on the growth of *P. falciparum*. We analyzed the complete 48-h life cycle of *P. falciparum* by Giemsa staining at 6 hpi, 18 hpi, 30 hpi, 42 hpi, and 54 hpi time points, and conducted the growth assay of *P. falciparum* at 5% oxygen (Fig. 4C–E) and 0.2% oxygen (Fig. 4F–H). Parasite growth at 5% O2 concentration was halted at the ring stage, unlike BEL-A WT reticulocytes that progressed from trophozoite to schizont stage during their 48-h life cycle. (Fig. 4C, D). Furthermore, our data demonstrated almost 50% reduction in invasion in BEL-A SCM compared to BEL-A WT at 6 hpi (Fig. 4D). The growth assay of *P. falciparum* at 0.2% oxygen condition demonstrated a comparable trend to that observed under 5% oxygen condition. Further, parasite invasion efficiency was reduced by almost 60% at 6 hpi under 0.2% O2 concentration (Fig. 4F, G). As a control, invasion and growth assay were performed with WT and SCD RBCs under both 5% O2 (Fig. 4C, E) and 0.2% O2 condition (Fig. 4F, H). The results showed a comparable growth pattern to that observed in BEL-A WT and BEL-A SCM reticulocytes. Overall, our data establishes the use of reticulocytes derived from pathologically relevant immortalized erythroid progenitors as a powerful in vitro SCD model system.

## Treatment of Hydroxyurea and Pomalidomide induces HbF expression in BEL-A SCM and BEL-A BTM-derived erythroblasts

In order to ascertain our disease lines are suitable cellular systems for screening small molecule drugs for SCD and BT, we conducted

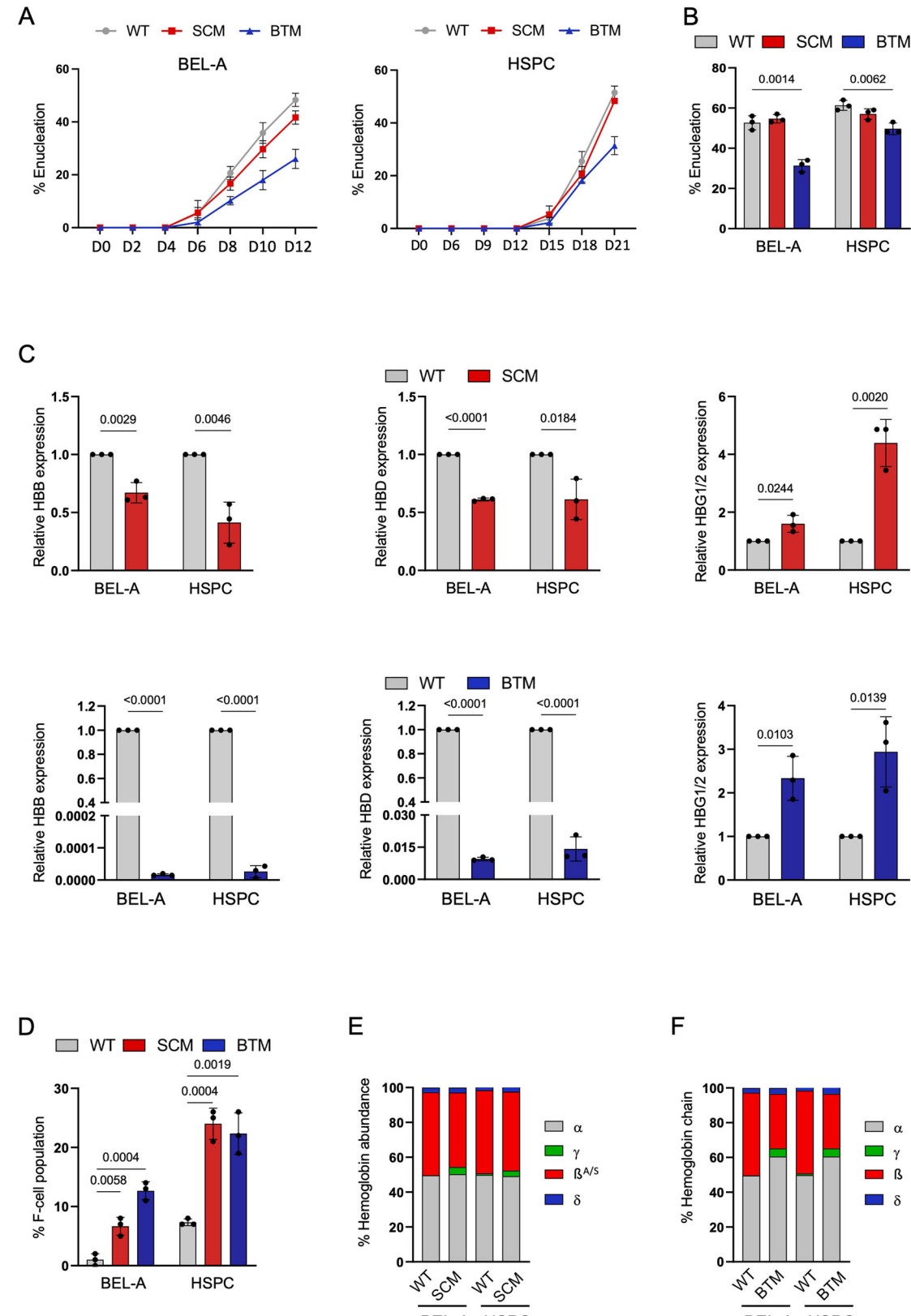

treatment studies with two known HbF inducing compounds hydroxyurea (HU) and pomalidomide (Pom). BEL-A WT, BEL-A SCM and BEL-A BTM cells were supplemented with HU (50 μM)[17] and Pom (5 μM)[18] and the results showed that BEL-A SCM had no significant difference in erythropoiesis and enucleation while improved enucleation by 19% and 20% was observed in BEL-A BTM treated with HU and Pom,

respectively (Fig. 5A). At the transcriptional level, increased γ-globin levels were observed in both BEL-A SCM and BEL-A BTM when treated with HU and Pom (Fig. 5B). Further, we also observed 17% and 25% increase in F-cell population in BEL-A SCM cells and 13% and 28% increase in BEL-A BTM cells treated with HU and Pom compared with untreated controls (Fig. 5C). Reactivated HbF was also detected in the

**Fig. 2 | Characterization of disease model cell lines, BEL-A SCM and BEL-A BTM.**
**A** Enucleation presented as percentage of Hoechst 33342 negative cells in flow cytometry during the progression of differentiation till Day 12 and Day 21 of BEL-A cells (WT, SCM, BTM) and HSPCs (WT, SCM, BTM). **B** Percentage of enucleation in BEL-A cells and HSPCs at Day 12 and Day 21 of differentiation and comparisons of enucleation efficiency of disease cells with respective WT. **C** Analysis of relative mRNA levels of *HBB, HBG, HBD* genes at day 6 of differentiation. The expression of the genes was normalized with the GAPDH housekeeping gene. The graph is plotted as mRNA levels of β-globin genes divided by α-gene levels. **D** Population of HbF antibody-stained positive cells in flow cytometry plots presented as percentage of F cell population. Percentage was normalized with its respective unstained controls.; **E**, **F** RP-HPLC analysis of globin chains was done at Day 10 and Day 21 of differentiation in (**E**) BEL-A SCM and HSPC SCM (**F**) BEL-A BTM and HSPC BTM respectively. All data is shown as Mean ± S.D from three independent replicates ($n = 3$). Statistical significance was determined by using two tailed student's *t*-test. Source data is provided in the Source file.

drug treated cells analyzed using RP-HPLC (Fig. 5D). Further, BEL-A WT cells treated with HU and Pom showed increased γ-globin expression at both transcriptional and protein levels (Supplementary Fig. 12). Next, we checked the rescue of ROS levels in BEL-A SCM and BTM cells treated with both the drugs. We found a reduction of 7.6% and 8.6% in BEL-A SCM and 17% and 17.33% in BEL-A BTM cells treated with HU and Pom, respectively (Fig. 5E). This suggests that treatment of the drugs alleviated cellular/disease physiology by reducing ROS levels. Further, flow cytometry analysis revealed a rescue of globin chain imbalance in BEL-A BTM by 51% and 45% (Fig. 5F), and RP-HPLC data demonstrated the rescue by 40% and 55% when treated with HU and Pom, respectively (Fig. 5G). Subsequently, we investigated whether these drugs could alleviate the sickling phenotype. Treatment of HU and Pom in BEL-A SCM led to a decrease of 26% and 23% in $β^S$ levels (Fig. 5H). The sickling assay further demonstrates a reduction of 31% and 27% in the number of sickled/abnormal reticulocytes. (Fig. 5I,J). Taken together, these results indicated that BEL-A SCM and BTM cells can serve as ideal cellular model systems with disease phenotype for screening and evaluating drug molecules of therapeutic potential.

### Generation of HPFH3 genotype significantly enhanced HbF expression in BEL-A SCM and BEL-A BTM derived erythroblasts

Next, to test whether our disease lines are amenable for genetic modifications, we introduced a naturally occurring deletional HPFH genotype using CRISPR-Cas9 in BEL-A SCM and BEL-A BTM. HPFH3 genotype is a naturally occuring deletion of a ~ 50 kb region of β-globin gene cluster on chromosome 11[19]. To evaluate the γ-globin induction potential of the HPFH3 genotype, a pair of sgRNAs targeting the 5′ and 3′ end of the HPFH3 were designed (Fig. 6A). HPFH3 genotype was generated in BEL-A WT, BEL-A SCM and BEL-A BTM lines with deletion efficiency of 46%, 52% and 49% respectively (Fig. 6B, Supplementary Figs. 13, 14A). Generation of HPFH3 genotype did not measurably alter the enucleation rate in BEL-A SCM edited cells however, the enucleation rate was increased in BEL-A BTM-edited cells (46%) compared with the unedited control (28%) (Fig. 6C). A 3.7-fold and 2.3-fold increase in the γ-globin mRNA levels was observed in HPFH3-edited BEL-A SCM, and BEL-A BTM edited cells, respectively (Fig. 6D). The cells with HPFH3 genotype showed a substantial increase in the proportion of F-cells, with percentages of 47%, and 49% in BEL-A SCM, and BEL-A BTM edited cells, respectively (Fig. 6E). HPFH3 genotype in BEL-A WT also demonstrated increased γ-globin levels at both mRNA and protein level (Supplementary Fig. 14). Further, RP-HPLC analysis demonstrated an increase in γ-globin in the HPFH3 edited cells (Fig. 6F), which led to a significant reduction in oxidative stress by 40% and 54% in BEL-A SCM and BEL-A BTM edited cells, respectively, thereby decreasing the generation of ROS (Fig. 6G). Flow cytometry and RP-HPLC demonstrates the restoration of globin chain imbalance in BEL-A BTM edited cells as compared to its unedited control (Fig. 6H,I). Additionally, the data indicates a concomitant reduction in HbS levels by 36% in HPFH3-edited BEL-A SCM cells (Fig. 6J). The proportion of sickle cells was significantly reduced in reticulocytes derived from HPFH3-edited BEL-A SCM cells with only 10% compared to 66% in unedited control (Fig. 6K, L). The results provide evidence of the ability of the HPFH3 genotype to induce HbF and

effectively rescue the disease-specific features in edited BEL-A disease models.

### CRISPR-mediated editing of *HBG1/2* gene promoter motif reactivates HbF

We further demonstrated the HbF reactivation potential of these cellular systems by editing the *HBG1/2* gene promoter motif in the β-globin gene cluster. We disrupted the *HBG1/2* gene promoter motif in BEL-A WT, BEL-A SCM and BEL-A BTM lines with the editing efficiency of 35%, 55% and 56% respectively (Fig. 7A and Supplementary Fig. 15). Non-homologous end joining (NHEJ)-mediated disruption of transcriptional repressors binding site in BEL-A SCM did not alter the erythroid differentiation and enucleation rate (Fig. 7B), however, the edited BEL-A BTM cells rescued the inefficient enucleation (Fig. 7B). We further observed a 4.0-fold, and 2.3-fold increase in γ-globin mRNA levels in edited BEL-A SCM, and BEL-A BTM cells, respectively (Fig. 7C). Editing also led to a 27%, and 25% increase in the F-cell population in BEL-A SCM, and BEL-A BTM cells, respectively (Fig. 7D), which was confirmed by RP-HPLC (Fig. 7E). A similar increase in γ-globin levels were found in BEL-A WT edited cells (Supplementary Fig. 16). The increased expression of γ-globin consequently resulted in 50% reduction in ROS levels in both BEL-A SCM and BTM edited cells compared to their unedited controls (Fig. 7F). Flow cytometry and RP-HPLC revealed a significant increase in total β-globin levels in edited BEL-A BTM cells (Fig. 7G,H). The edited BEL-A SCM-derived erythroblasts showed 57% reduction in HbS (Fig. 7I) leading to reduced number of sickle cells (32%) due to reactivated HbF, as compared to unedited controls (63%) (Fig. 7J,K). These results suggest that the targeted site on the *HBG1/2* promoter is crucial for gene regulation and its disruption can successfully mitigate the disease-specific features in edited BEL-A disease models.

## Discussion

The development of immortalized erythroid progenitors using CRISPR technology to create isogenic cell lines for SCD and BT has been a highly sought-after goal in the field of erythroid biology. Despite several studies generating hiPSC lines from SCD and BT patients[20–23] no in vitro cellular system could produce reticulocytes equivalent to primary adult reticulocytes for these two blood disorders[24]. Recently, an immortalized erythroid line siBBE, was generated from peripheral blood stem cells of a HBE/β-thalassemia patient[25], but the pathophysiological characteristics of this line were not directly compared with primary patient-derived cells.

To overcome these challenges, we generated BEL-A SCM and BEL-A BTM lines that underwent terminal erythroid differentiation and produced mature enucleated erythrocytes. Furthermore, through extensive molecular analysis and comparison with patient specific HSPC-derived erythroblasts, we found that erythroblasts derived from BEL-A SCM and BEL-A BTM accurately replicated the disease phenotype at the molecular and cellular level. It is important to note that the fold change of γ-globin is higher in erythroblasts derived from disease-specific HSPCs as compared to erythroblasts from disease–specific BEL-A cells. This increased γ-globin levels in disease-specific HSPCs could potentially be due to genetic modifiers present in patients from which the HSPCs have been isolated, highlighting the importance of

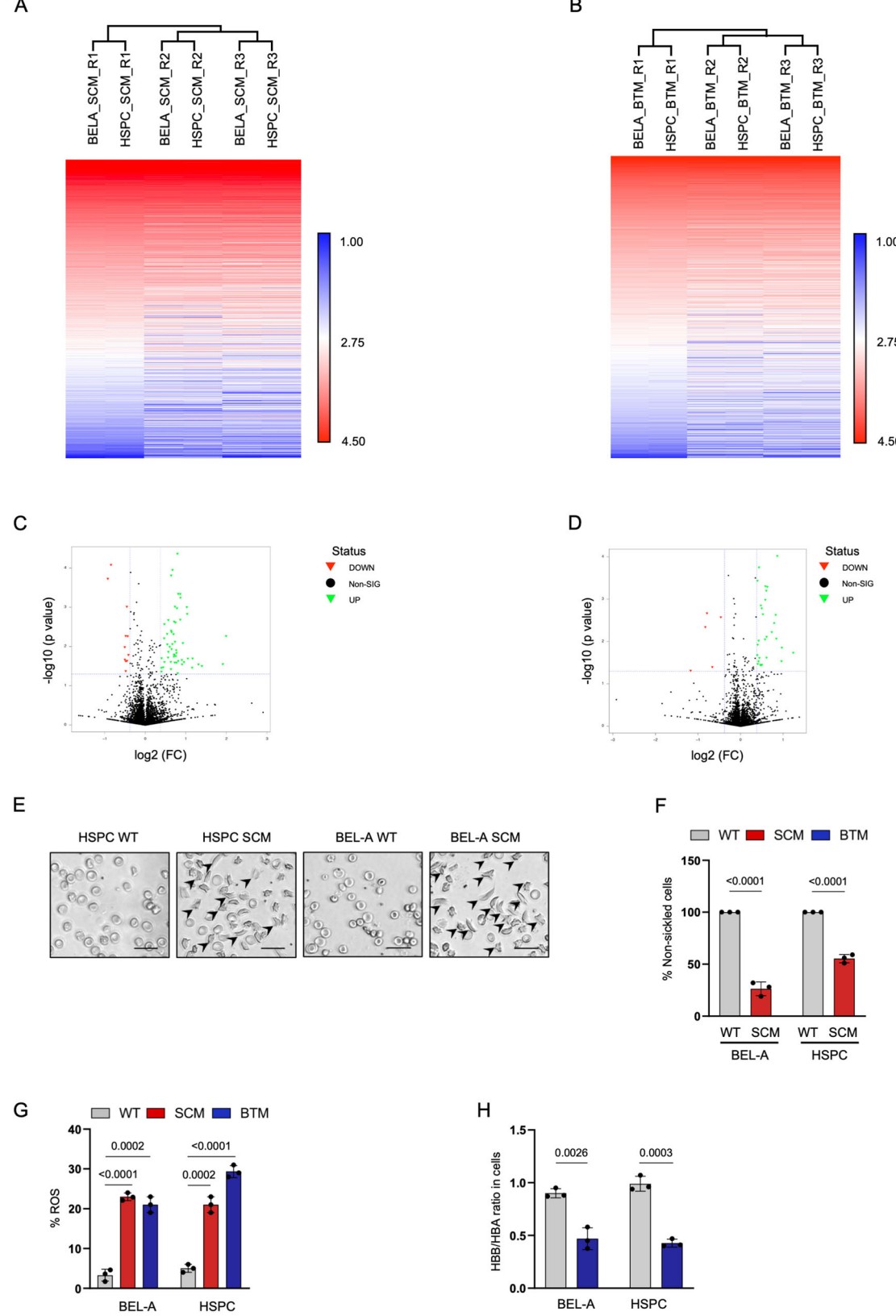

accounting for inter-individual variability when comparing cell populations from different genetic origins[26,27]. Overall, our molecular, phenotypic, and proteomic data validate the disease specificity of BEL-A SCM and BEL-A BTM cells for all parameters and confirm that our disease lines recapitulate the key aspects of respective disease pathology, such as sickling under hypoxia conditions, globin chain imbalance, and increased ROS production. A similar finding was reported very recently through the generation of cellular model systems for BT[28].

Our study demonstrates that reticulocytes derived from BEL-A SCM exhibit a similar level of resistance to *P. falciparum* invasion as SCD patient-derived RBCs. Numerous mechanisms including sickling

**Fig. 3 | BEL-A SCM and BEL-A BTM are similar to disease specific HSPCs and recapitulate disease physiology. A**, **B** Heatmaps of proteomic data from (**A**) BEL-A SCM with SCM HSPCs and (**B**) BEL-A BTM with BTM HSPCs presented as log 10 normalized data. Red indicates high protein expression and blue indicates low protein expression. **C**, **D** Volcano plots of (**C**) BEL-A SCM with SCM HSPCs and (**D**) BEL-A BTM and BTM HSPCs are presented as -log10 *p*-value and -log2 fold change value. Upregulated proteins (fold change of ≥1.5) are depicted in green and downregulated proteins (fold change of ≤0.67) are depicted in red with *p*-values < 0.05. Quantile normalization was done and statistical significance was determined by two tailed *t*-test. **E** Representative micrographs of sickling assay done in BEL-A WT, BEL-A SCM, WT HSPC and SCM HSPC derived reticulocytes.

Scale: 20 μm (**F**) Quantification of sickled cells in BEL-A WT, BEL-A SCM, WT HSCPs and SCM HSPCs derived reticulocytes. Percentage sickled cells presented as number of cells with sickle and abnormal morphology by total number of cells (≥200 cell counts). **G** Percentage of ROS in BEL-A (WT, SCM and BTM) and HSPCs (WT, SCM and BTM). Data is plotted as percentage positive cells in flow cytometric analysis. **H** Flow-cytometric analysis of β-like globin by α-globin in BEL-A (WT and BTM) and HSPCs (WT and BTM) derived erythroblasts. All experiments were done in triplicates independently (*n* = 3) and data is presented as Mean ± S.D. Statistical significance was determined by using two tailed student's *t*-test. Source data is provided in the Source file.

of infected RBCs, translocation of HbS-specific parasite-growth inhibiting miRNAs, stimulation of heme-oxygenase-1 and lower oxygen states are plausible explanations for HbSS malaria resistance[29]. Despite decades of research, the precise biological mechanism through which the HbSS mutation protects against severe malaria remains elusive[30,31]. BEL-A SCM-derived sickled reticulocytes offer a continuous and renewable supply for targeted exploration of host-receptor involvement in malarial pathogenesis. LaMonte et al demonstrated that host-specific miRNAs expressed in HbSS reticulocytes can inhibit the translation of parasite enzymes essential for parasite growth and development[32]. Compared to SCD patient HSPCs, BEL-A SCM cells are more amenable to manipulation at the genetic level and have an unlimited proliferative capacity, providing an advantage in investigating the relevant cellular and molecular mechanisms underlying the protective effects of the HbSS mutation.

Large scale drug screening is an essential avenue for exploring drug molecules for the management of SCD and BT but is significantly hindered by the restricted renewal capacity and limited access to patient-derived primary HSPCs. Nonetheless, leukemic cell lines and erythroleukemic cell line such as K562 have been used for screening, but they do not accurately reflect erythropoiesis[33]. In addition, earlier reports have shown that widely studied HUDEP-2-derived erythroid cells do not respond to the treatment of different drug molecules known to enhance the γ-globin expression in adult erythrocytes[7]. Data from our engineered pathophysiologically relevant cellular systems suggest their potential as a drug screening platform. The addition of HU as a HbF enhancer for SCD patients and pomalidomide-treated cells showed significantly enhanced expression of γ-globin than untreated controls at mRNA and protein levels, similar to that found in immortalized lines derived from healthy donor cells[15,17,25] and concomitant reduction in sickle phenotype and ROS levels in both disease cells. Studies have raised concerns regarding the safety profile of HU due to its associations with thrombocytopenia, myelotoxicity, and higher infection rates[34,35]. Moreover, HU-induced HbF is not uniformly distributed in all erythroblasts, hence it is only partially effective[36,37]. Consequently, continuous efforts are needed to develop inexpensive oral drugs for SCD and BT patients.

Clinical symptoms of SCD and BT are alleviated in patients who co-inherit the HPFH genotype[38]. Here, we report a proof of concept of their ex vivo genetic manipulation potential by generating naturally-occurring beneficial HPFH3 genotype[13] in the β-globin gene-cluster using genome-editing for HbF induction. The heterozygosity of HPFH3 genotype leads to 25% to 30% of HbF expression in pancellular manner[39], which has therapeutic potential for β-hemoglobinopathies[40]. Moreover, HPFH3 deletion is not associated with any clinical abnormalities[39]. Our results demonstrated that HPFH3-edited erythroid progenitors produced erythroblasts with therapeutically relevant levels of HbF that were sufficient to reverse the phenotypes of BT and SCD. Enhancement of HbF in HPFH3-edited BEL-A BTM cells balances the α-globin/β-like globin chain ratio, resulting in improved erythroid lineage maturation and decreased ROS levels. Studies have reported that a reduction in ROS levels in the thalassemic mouse models has shown to improve their clinical profile[41,42]. Our findings support the hypothesis

that the HPFH3 genotype which removes ~50 kb region in the β-globin gene cluster results in increased HbF due to following reasons contributing independently (1) the juxtaposition of 3′ HS1 to *HBG1/2* genes[43], (2) deletion of competitor genes; *HBD* and *HBB*, (3) deletion of putative negative regulators between the *HBG1* and *HBD* genes[44,45]. Our detailed evaluation furnished the proof-of-principle demonstration of recapitulation of the HPFH3 genotype for the induction of HbF using CRISPR-mediated genome-editing, which could realize the therapeutic promise of this beneficial genotype.

In addition, we also demonstrate that site-specific disruption of region 118 to 114 nucleotides upstream of the transcription start sites (TSS) of the identical *HBG1/2* genes in BEL-A SCM and BEL-A BTM cells reactivated the γ-globin expression and reversed the disease conditions, consistent with earlier observations[46]. Targeted editing of this region may disrupt binding sites for several transcriptional repressor proteins such as nuclear factor Y (NF-Y)[47], CCAAT displacement protein (CDP)[48] and BCL11A[49]. Our findings highlight the advantages of these physiologically relevant cellular systems with their amenability to genetic manipulations.

In summary, we present a major step towards the development of pathophysiologically relevant cellular systems for SCD and BT for identifying and investigating host-receptors for malaria invasion, novel regulators of γ-globin expression and a wider range of applications. Here, we further demonstrate proof of concept of their ex vivo research potential by creating HPFH3 genotype and *HBG1/2* promoter motif editing to reactivate HbF for the treatment of β-hemoglobin disorders.

## Methods

### CD34+ Hematopoietic stem progenitor cell (HSPCs) culture
Peripheral blood from healthy donors, BT, and SCD patients eligible for research purposes were obtained under written informed consent. The samples obtained from human donors were de-identified and no information was available on the sex of the donors. The study was approved by the Institutional Human Ethics Committees at Institute of Genomics and Integrative Biology and Thalassemia and sickle cell society (CSIR-IGIB/IHEC/2017-18/12; TSCS-1112018; 2020-001-EMP-28) and used according to the Declaration of Helsinki. The blood samples were subjected to peripheral blood mononuclear cells (PBMC) isolation by the Ficoll gradient method[50] followed by CD34+ cells purification. The purification was performed using EasySep™ Human CD34 positive selection kit II (Stem cell Technologies) with the help of EasySep™ Magnet (STEMCELL Technologies) as per the manufacturer's instructions. The purified CD34+ HSPCs were cultured in StemSpan™ SFEM II medium supplemented with CD34+ Expansion cytokines (STEMCELL Technologies) for maintenance and expansion. The cells were cultured for 6 days followed by erythroid differentiation.

### sgRNA design and validation
Single guide RNAs (sgRNAs) were designed using CRISPR/cas9 guide RNA design checker (IDT) and CHOPCHOP[51]. The most efficient sgRNAs with high specificity and the least off-target effects were

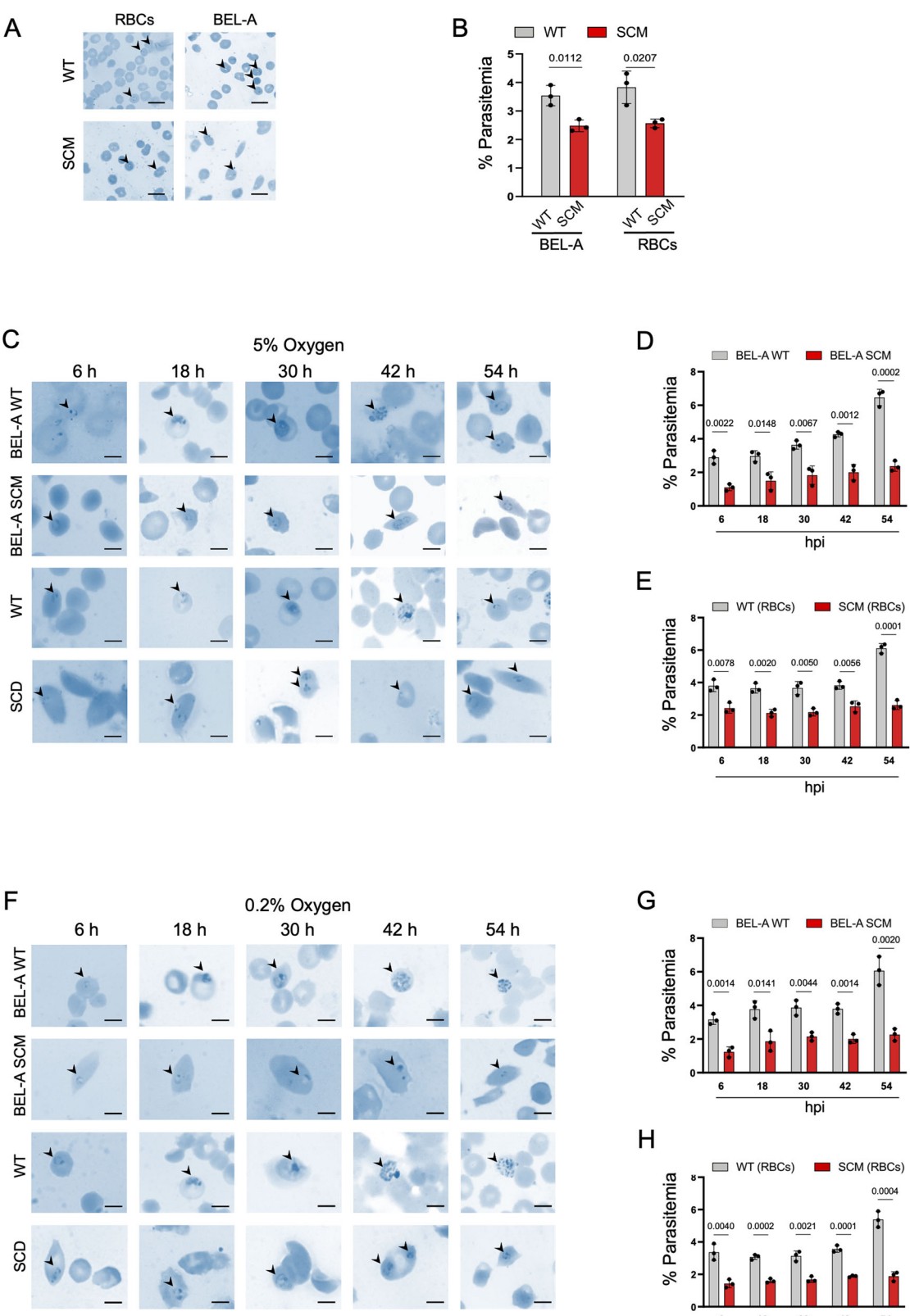

chosen. All the sgRNAs were cloned in pSpCas9(BB)-2A-Puro(pX459) (Addgene #62988) and the activity of sgRNAs at the endogenous target site was initially validated in HEK293T cells using T7 endonuclease I assay[52]. sgRNA sequences, targets and genotyping primers used in this study are given in the Supplementary Tables 1, 2.

## Construction of piggyBac-based donor plasmid

piggyBac DONOR constructs, pDONOR-tagBFP-PSM-EGFP (Addgene #100603) and pDONOR-tagBFP-PSM-dTOMATO (Addgene #100604) were used to introduce sickle cell mutation (SCM) in both alleles. Homology arms were amplified from sickle cell patient-derived

**Fig. 4 | BEL-A SCM shows physiologically similar invasion efficiency as SCD patients derived RBCs. A** Representative Giemsa microscopic images of parasite invasion. WT RBCs and BEL-A WT reticulocytes showed normal merozoite invasion at 12 h post infection (hpi); while SCD erythrocytes and BEL-A SCM reticulocytes showed reduced invasion. Arrows indicate infected RBCs (Red Blood cells) successfully invaded by merozoites. **B** Quantification of parasite invasion by plotting percentage parasitemia. Data is presented as Mean ± S.D. **C** Representative microscopic images of Giemsa-stained smears at time points 6 hpi, 18 hpi, 30 hpi, 42 hpi and 54hpi for the parasite invasion assay done at 5% O2 condition and its (**D, E**) Quantification of parasite invasion efficiency by plotting percentage parasitemia. Data is presented as Mean ± S.D. **F** Representative microscopic images of Giemsa-stained smears at time points 6 hpi, 18 hpi, 30 hpi, 42 hpi and 54 hpi for the parasite invasion assay done at 0.2% O2 condition and its (**G, H**) Quantification (Mean ± S.D) of parasite invasion efficiency. All experiments were done in three independent replicates ($n = 3$). Scale bar: 10 μm. Statistical significance was determined by using two tailed student's $t$-test. Source data is provided in the Source file.

genomic DNA and assembled using Gibson isothermal assembly kit (NEB, USA) as per the manufacturer's instructions on the above mentioned pDONOR constructs, resulting in the final donor vectors, pDONOR-eGFP-SCM and pDONOR-dTomato-SCM. Similarly, to introduce β-thalassemia IVS1-5 mutation (BTM), the homology arms were amplified from a β-thalassemia patient genomic DNA and assembled using the Gibson isothermal method in the piggyBac DONOR constructs. The resultant plasmids were named pDONOR-eGFP-BTM and pDONOR-dTomato-BTM. For ease of screening, a silent mutation was introduced in the right homology arm which generated a new HhaI restriction site. The complete sequences of homology arms and primers for PCR amplification used are provided in Supplementary Tables 3, 4 respectively.

## BEL-A culture
BEL-A cells, an immortalized erythroid progenitor cell line (a gift from Profs. Jan Frayne, David Anstee and Dr. Kangtana Trakarsanga, University of Bristol, UK) were cultured[11] in serum-free expansion medium (StemSpan SFEM II®, StemCell Technologies) supplemented with 1x Antibiotic Antimycotic solution (Sigma Aldrich), 50 ng/mL human recombinant Stem Cell Factor (Immunotools), 3 IU/ mL Erythropoietin (Peprotech), 1 μg mL Doxycycline (Sigma Aldrich), and $10^{-6}$ M Dexamethasone (Sigma Aldrich). The cells were cultured at 37 °C in a 5% CO2 incubator.

## Generation of BEL-A SCM and BEL-A BTM erythroid progenitor lines
The disease-specific isogenic lines, BEL-A SCM and BEL-A BTM, were generated from a well-characterized erythroid progenitor cell line BEL-A[11] using piggyBac transposase system integrated with CRISPR-mediated genome- editing. $0.5 \times 10^6$ BEL-A cells were electroporated with a total of 500 ng plasmid in a ratio of 1:2:2 (pX459-HBBsgRNA: pDONOR-eGFP-SCM or pDONOR-eGFP-BTM: pDONOR-dTomato-SCM or pDONOR-dTomato-BTM) at 1100 V, 30 ms, 3 pulses using the Neon Transfection system (Thermo Fisher Scientific) as per the manufacturer's protocol. Post nucleofection, when the BFP signal could no longer be detected by flow cytometry, puromycin selection (1 μg/ml) was initiated for the next 72 h. Subsequently, to obtain an enriched subset of potentially biallelic gene-edited cells, eGFP+dTomato+ BFP- cells were sorted. The cell sorting was done in a hierarchical manner, firstly, FSC-A/SSC-A gating was applied to exclude cell debris and acquire target population of cells followed by gating of cells in FSC-W/FSC-H and SSC-W/SS-H to exclude doublets. Subsequently, BFP- cells were gated and amongst them eGFP+ and dTomato+ cells were selected and sorted in BD FACSAriaIII. eGFP+dTomato+BFP- cells were allowed to amplify and were single cell sorted and expanded. On day 15, single cell sorted clones were screened by Flow cytometry, PCR and Sanger sequencing.

## Removal of selection cassette using piggyBac transposon
Biallelically targeted clones were used for transposon removal. Transposase treatment was given to BEL-A SCM$^{eGFP+dTomato+}$ and BEL-A BTM$^{eGFP+dTomato+}$ mutant lines to remove the selection cassette (positive selection module; PSM) from the integration site. $0.5 \times 10^6$ BEL-A mutant cells were mixed with 500 ng of piggyBac transpose expression plasmid (System Biosciences) and electroporated using a Neon

electroporation device as described above. The cells were then allowed to amplify. Two weeks post nucleofection dual negative, i.e., SCM$^{eGFP-dTomato-}$and BTM$^{eGFP-dTomato-}$ cells were sorted separately as one cell/well in 96 well plates.

## Genome-editing mediated generation of HPFH beneficial genotypes
CRISPR/Cas9 ribonucleoprotein (RNP) complex was formed by combining 1.5 μg of Cas9 protein (CAS9GFPPRO; Sigma) with 0.5 μg of chemically modified guide RNA (2′-O-methyl analogs and 3′ phosporothioate modifications were included at the three terminal nucleotides of the 5′ and 3′ ends; Sigma) for 10 min at room temperature. For *HBG1/2* gene promoter editing, we used 1.5 μg of Cas9 with 0.5 μg of chemically modified sgRNA. For dual sgRNA-mediated HPFH3 genome editing, we used 1.5 μg of Cas9 with 0.25 μg of chemically modified sgRNA for target site-I sgRNA and 0.25 μg of chemically modified sgRNA for target site-II sgRNA. $0.5 \times 10^6$ BEL-A WT, BEL-A SCM and BEL-A BTM cells were electroporated using a Neon electroporation device as described above. The electroporated cells were immediately neutralized with a pre-warmed culture medium with cytokines and allowed to recover for 48 h before proceeding to erythroid differentiation.

## ddPCR-based quantification of genome editing
The deletion efficiency of the HPFH3 edited BEL-A cells using CRISPR/Cas9 was determined through a ddPCR-based analysis. PCR amplification was done using two sets of primers: (1) the deletion-specific 5BP-F with 3BP-R primers, and (2) the unedited/wildtype-specific 5BP-F with 5BP-R and 3BP-F with 3BP-R primers. The primer information can be found in Supplementary Table 5. The amplified products were further subjected to amplification with EvaGreen® Supermix(BioRad) and the above set of primers using the QX200 BioRad System. The deletion efficiency was calculated using the formula provided.

$$Deletion\ Efficiency = ((D)/(D + W)) \times 100$$

where,
 D is Read count of deletion-specific amplicon
 W is Read count of unedited or wildtype specific amplicon

## Amplicon sequencing
The editing events and efficiency of the *HBG1/2* promoter edited samples were analysed using amplicon based next generation sequencing method[53]. The PCR amplicons were subjected to fragmentation followed by tagging to adapter sequences (tagmentation). Another round of PCR was performed on the adapter tagged fragments, the adapter index tagged libraries were pooled and cleaned using the purification beads. High Sensitivity dsDNA quantification kit (Invitrogen) was used to quantify the final library. Post normalization and quality assessment the synthesized library was subjected to the paired-end sequencing on Illumina sequencing platform (Illumina Inc.)

For analysis the Fastq files were quality trimmed using Trimmomatic and aligned using bwa mem, further the duplicate reads were removed using picard MarkDuplicates. The resultant BAM file is then analyzed using the R package CrispRVariants for identifying the editing events and efficiency for each sample (Picard, https://broadinstitute.github.io/picard)[54–56].

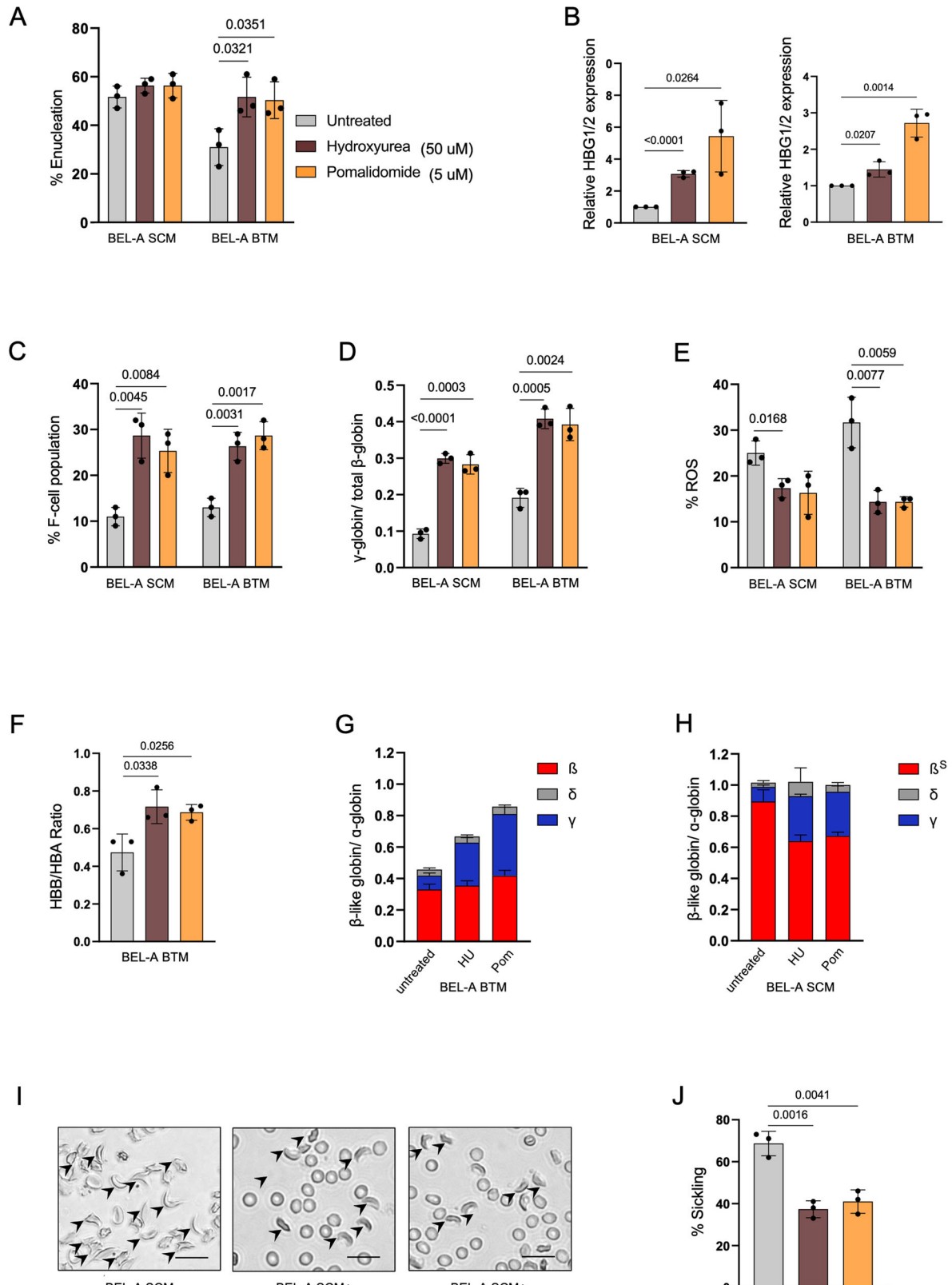

## Erythroid differentiation of BEL-A cells

The BEL-A WT, disease BEL-A lines and edited BEL-A cells were differentiated into erythroid cells by culturing with differentiation medium of IMDM (Sigma) supplemented with 3% human AB serum (Sigma Aldrich), 2% Fetal Bovine Serum (FBS) (Thermo Fisher Scientific), 1X Antibiotic Antimycotic solution (Sigma Aldrich), 200 μg/mL Holo-transferrin (Sigma Aldrich), 10 μg/mL recombinant human insulin (Sigma Aldrich), 3 IU/mL heparin (STEMCELL-Technologies), 3 IU/mL erythropoietin (Peprotech), 1 μg/mL doxycycline (Sigma Aldrich), 10 ng/mL stem cell factor (Immunotools),

**Fig. 5 | BEL-A SCM and BEL-A BTM are therapeutically relevant disease models for drug validation. A** Percentage of enucleation after drug treatment. **B**–**D** Estimation of increase in γ-globin gene expression. **B** Analysis of relative *HBG1/2* gene expression after treatment in BEL-A SCM and BEL-A BTM at Day 6 of differentiation by qRT PCR. **C** Flow cytometry-based quantification of F-cell population after drug treatment in BEL-A SCM and BEL-A BTM cells. **D** RP-HPLC of globin chains. Data is presented as the abundance of γ globin by abundance of total β- like goblins (β + γ + δ). **E**–**J** Determination of rescue in disease phenotype and/or physiology at day 10 of differentiation (**E**) Percentage of ROS in BEL-A SCM and BEL-A BTM. **F** Flow-cytometric analysis of β-like globin divided by α-globin in BEL-A BTM differentiated cells. **G, H** RP-HPLC plotted as β-like globins (β + γ + δ) divided by α-globin in (**G**) BEL-A BTM differentiated cells and (**H**) BEL-A SCM differentiated cells treated with HU (hydroxyurea) and Pom (Pomalidomide). **I** Representative microscopic images of sickling in BEL-A differentiated cells at Day 12 (≥200 cell counts), Scale bar: 20 μm and its (**J**) quantification. All experiments were done in triplicates independently (*n* = 3) and data is presented as Mean ± S.D. Statistical significance was determined by using two tailed student's *t*-test. Source data is provided in the Source file.

1 ng/mL Interleukin-3 (IL-3) (Immunotools). The cells were seeded at $2 \times 10^5$ cells/mL density at day 0 followed by $3.5 \times 10^5$ cells/mL on day 2. The cells were further seeded at $5 \times 10^5$ cells/mL on day 4 and doxycycline was removed from the medium. From day 6 onwards, $1 \times 10^6$ cells/mL density was maintained and the concentration of holo-transferrin was increased to 500 μg/mL. On day 6, SCF and IL-3 were also removed from the medium and the cell density was maintained at $1 \times 10^6$ henceforth until Day 12[11]. At the end of erythroid differentiation, erythroblasts were isolated by centrifuging at 150 *g* for 5 min.

### Erythroid differentiation of CD34+ HSPCs
CD34+ HSPCs were differentiated into mature erythroid cells using the three-phase erythroid differentiation protocol[57]. During phase 1 (day 0 to day 8), $5 \times 10^4$ cells were cultured in an erythroid differentiation medium (EDM) consisting of IMDM (Sigma) supplemented with 100 ng/ml recombinant human SCF, 330 μg/ml human holo-transferrin, 5 ng/ml human IL-3, 3 IU/ml recombinant human erythropoietin (EPO), 2 IU/ml heparin, 5% human AB serum, 20 μg/ml Insulin, 1% GlutaMAX supplement and 1X Antibiotic Antimycotic. During phase 2 (day 9 to 14), cells were seeded in a density of $2 \times 10^5$ and cultured in EDM medium without IL-3. During phase 3 (Day 14–21), SCF was also removed from EDM, and cells were cultured at a density of $5 \times 10^5$ cells/ml thereafter.

### Drug treatment of BEL-A SCM and BEL-A BTM lines
The cells were pre-conditioned by supplementing 50 μM hydroxyurea (H8627; Sigma) and 5 μM Pomalidomide (P0018; Sigma) in BEL-A expansion medium for 4 days. The cells were differentiated as described above, and the drugs were supplemented at the same concentration until day 6. Subsequently, the drugs were removed and differentiation was continued until day 12[17].

### Flow cytometry analysis for erythroblast characterization
$1–3 \times 10^5$ cells were harvested on day 12 for BEL-A and Day 21 for HSPCs of erythroid differentiation and stained for differentiation markers anti-CD71-PE (1:100, 60106PE, Clone OKT9, Stem Cell Technologies) and Glycophorin-A-FITC (1:50, 60152FI, clone 2B7, Stem Cell technologies). The cells were incubated with antibodies for 20 min followed by washing with PBS and flow cytometry analysis

For evaluating the efficiency of enucleation, cell-permeable double-stranded DNA dye Hoechst 33342 (2 μg/ml, R37165, Thermo Fisher Scientific) was used according to the manufacturer's instructions. Briefly, the cells were stained with 2 μg/ml Hoechst 33342 and incubated for 20 min at room temperature. This was followed by washing the cells with PBS and analysis by flow cytometry.

For evaluating the percentage of hemoglobin variants expressed in the total population, $1–3 \times 10^5$ erythroid differentiated cells were fixed with 4% formaldehyde (Sigma Aldrich) for 10 min, permeabilized with 0.1% Triton X-100 (Sigma Aldrich) for 5 min. Cells were then washed with PBS supplemented with 2% FBS and stained with anti-HbF-1 (1:50, MHFH05, Thermo Fisher Scientific), anti-hemoglobin β antibody (1:50, sc-21757 FITC, clone 37-8, Santa Cruz Biotechnology) and Anti-hemoglobin α antibody (1:50, sc-514378 PE, clone D-4, Santa Cruz

biotechnology). Cells were then washed with PBS and analyzed by flow cytometry.

To determine the reactive oxygen species (ROS) levels, $5 \times 10^5$ cells were stained with CM-H2DCFDA (1 μm, C6827, Thermo Fisher Scientific) for 30 min as per the manufacturer's instructions. Subsequently, the stained cells were washed with PBS and analyzed using flow cytometry.

All the acquisitions were done in BD Accuri C6 analyzer and the data were analyzed in FlowJo software (Version 10). The details of the antibodies and dyes used in the study are compiled in Supplementary Table 6.

### Quantitative Real-Time PCR (qRT PCR)
$2 \times 10^5$ cells were harvested for total RNA isolation on day 7 of BEL-A and day 15 of CD34+ HSPCs erythroid differentiation. RNA was isolated using TRIzol™ (Thermo Fisher Scientific). Total RNA was quantified by spectrophotometer (Nanodrop 2000, Thermo Fisher Scientific). For reverse transcription, 1 μg of total RNA was used for the cDNA synthesis using high-capacity cDNA synthesis kit (Thermo Fisher Scientific) as per the manufacturer's protocol. The resultant cDNA was used as a template for qPCR with KAPA SYBR Fast (Kapa Biosystems) qRT PCR was done for following genes: *HBB, HBA, HBD, HBG, GAPDH, AHSP, BAND3, BCL2L, XPO1, SPTB1, EPOR*. In addition, cDNA was used as a template for semiquantitative PCR for erythroid lineage specific Transcription Factors including *EKLF, EKLF-3, PABPC1, LMO2, SOX-6, FOG-1, BCL11A, ZBTB7A, GATA-1*. The details of all the primers used in the study are mentioned in the Supplementary Table 7.

### Giemsa staining
On day 12 of differentiation, $1 \times 10^5$ cells were harvested, washed with PBS, resuspended in 30 μL PBS, and smeared onto a clean glass slide. The slide was air-dried, fixed in methanol for 2 min, and then stained with Giemsa (20% in milliQ water) for 5 min, followed by rinsing with milliQ water. The slide was then air-dried, and images were captured using a Brightfield microscope (EVOS M5000, Thermo Fisher Scientific) at 40X magnification.

### Reverse phase-high-performance liquid chromatography (RP-HPLC) analysis of Globin chains from human erythroid cells
Globin chain analysis was performed on a reverse phase HPLC system[58]. Briefly, $1 \times 10^6$ BEL-A cells were harvested at day 12 of differentiation and centrifuged at 200 × g for 5 min. Supernatant was discarded and pellet was lysed with the addition of 900 μl of ice-cold MilliQ followed by vortexing and incubation on ice for 20 min. This solution was centrifuged at 9500 x g for 10 min at 4 °C. Supernatant was diluted 1:100 with ice-cold MilliQ and loaded in HPLC vial. 20 μl of the sample was loaded in the column for separation where Buffer A (5% Acetonitrile, 0.1% trifluoroacetic in MilliQ deionized water) was used as loading buffer and Buffer B (95% Acetonitrile, 0.1% trifluoroacetic in MilliQ deionized water) was used as elution buffer.

### In vitro sickling assay
On day 12 of differentiation, the cells were incubated for 20 min to form loose red pellets. Supernatant was carefully removed. $1 \times 10^4$ cells

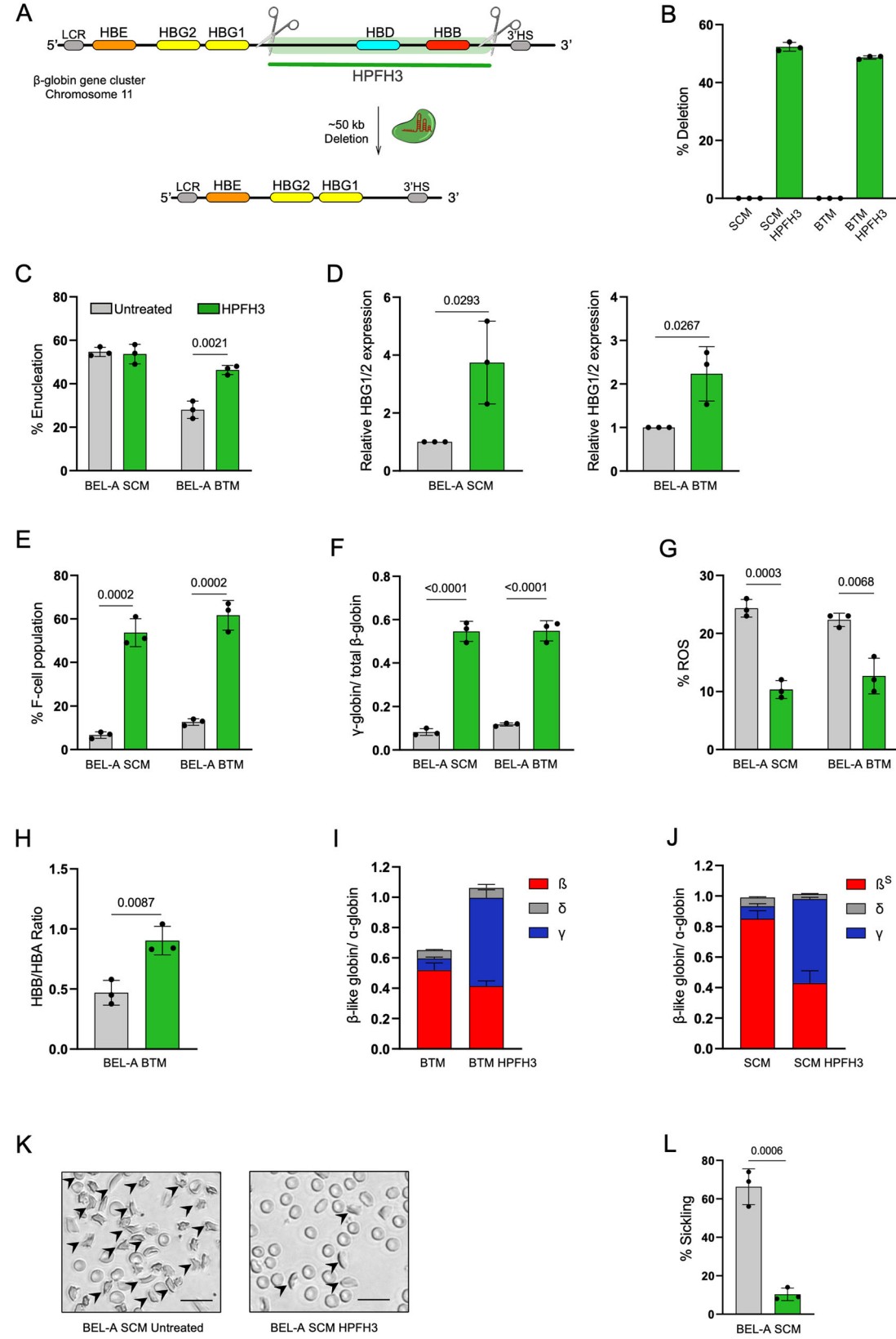

were resuspended in 150 μL HBSS buffer and seeded in a 96-well plate. The plate was then kept in an incubator (Eppendorf, New Brunswick Galaxy 48 R) at 0.2% oxygen at 37 °C and 5% CO2. After 4 h, BEL-A SCM and BEL-A WT cells derived reticulocytes were assessed for abnormal/ sickle shaped morphology using the Floid Cell imaging system at 20X magnification.

**Parasite culture**

*Plasmodium falciparum* 3D7 line was thawed and cultured in RPMI-HEPES medium supplemented with 2 g/L sodium bicarbonate (Sigma-Aldrich, USA), 5 g/L Albumax (Invitrogen, USA), 50 mg/L Hypoxanthine (Sigma-Aldrich, USA) and 10 μg/mL gentamicin sulfate (Invitrogen, USA) using human O+ erythrocytes. Culture was maintained at 2%

**Fig. 6 | CRISPR-mediated genome editing recapitulates HPFH3 genotype and rescues disease phenotype.** CRISPR/Cas9-mediated genome editing was utilized for generating HPFH3 deletion (Indian HPFH3). **A** Schematics showing target region for HPFH3 genotype recapitulation at β-globin cluster on Chromosome11. **B** Efficiency of HPFH3 deletion in BEL-A SCM and BEL-A BTM cells estimated using ddPCR. (**C**) Enucleation percentage of unedited and HPFH3-edited BEL-A SCM and BEL-A BTM cells. **D–F** Estimation of increase in γ-globin gene expression. (**D**) Relative *HBG1/2* gene expression of unedited and HPFH3-edited BEL-A SCM and BEL-A BTM cells. **E** Flow cytometry-based quantification of F-cell population of unedited and HPFH3 edited BEL-A SCM and BEL-A BTM cells. **F** RP-HPLC of globin chains in BEL-A SCM and BEL-A BTM HPFH3 edited cells. Data is presented as the abundance of γ-globin by abundance of total β-like globins (β + γ + δ).

**G–L** Determination of rescue in disease phenotype and/or physiology at day 10 of differentiation (**G**) Percentage of ROS in unedited and HPFH3 edited BEL-A SCM and BEL-A BTM cells. **H** Flow-cytometric analysis of β-like globins divided by α-globin in unedited and HPFH3 edited BEL-A BTM differentiated cells (**I, J**) RP-HPLC plotted as β-like globins (β + γ + δ) divided by α-globin in (**I**) BEL-A BTM unedited and HPFH3 edited differentiated cells and (**J**) BEL-A SCM unedited and HPFH3 edited differentiated cells. **K** Representative microscopic images of sickling in BEL-A SCM unedited and edited differentiated cells at Day 12 (≥200 cell counts), Scale bar:20 μm and its (**L**) quantification. All experiments were done in triplicates independently (*n* = 3) and data is presented as Mean ± S.D. Statistical significance was determined by using two tailed student's *t*-test. Source data is provided in the Source file.

hematocrit at 37 °C under mixed gas conditions (5% CO2, 5% O2, 90% N2).

## Invasion assay

Invasion assay was performed with reticulocytes derived from BEL-A SCM and BEL-WT as well as RBCs derived from SCD patients and healthy individuals. For this, *P. falciparum* 3D7 strain was purified using percoll gradient centrifugation (GE Healthcare) to enrich the 44–46 h schizont stage parasites with a purity of 95%. Purified schizonts were then incubated with the respective erythrocytes for the invasion assay in a culture volume of 100 μL in 96-well plate with the initial parasitemia and hematocrit of 1% and 2% respectively. After 10 h of incubation, slides were prepared and stained with Giemsa (Sigma-Aldrich, USA). Parasites in the ring stage were examined and counted under a light microscope.

## Parasite growth assay

To interrogate the effect of parasite growth in reticulocytes derived from BEL-A SCM and BEL-WT as well as RBCs derived from SCD patients and healthy individuals, parasite growth was monitored for a complete life cycle. For this, the parasite was percoll-purified and cultured in 96 wells to maintain the 1% parasitemia and 2% hematocrit with respective erythrocytes/reticulocytes. The development of the parasites was monitored at two different oxygen conditions. In one condition, the culture plate was incubated at 37 °C in a humidified culture chamber along with 0.2 % oxygen while in the other it was maintained at 5% oxygen. Parasites were monitored after every 10 h to examine the parasite growth for the course of one cycle.

## TMT labeling, mass spectrometry and data analysis

Sample preparation for proteomics analysis was performed[59] on BEL-A BTM, BEL-A SCM, HSPCs BTM, and HSPCs SCM cells. Lysis was done using a 2% lysis buffer (1X Halt Protease and Phosphatase inhibitor cocktail, 2% SDS and 50 mM TEAB) in cold conditions. The protein content in the samples was measured using the Pierce™ BCA Protein Assay kit as per the manufacturer's protocol. Following the BCA estimation, an SDS-PAGE analysis was conducted for quality control. Subsequently, 400 μg of protein was subjected to reduction and alkylation using 10 mM DTT and 20 mM IAA respectively. The protein was precipitated by adding 6X volume of chilled acetone and kept for overnight incubation at −20 °C. 400 μg of protein samples was digested using Trypsin (Promega) in the ratio 1:20 (trypsin: protein) and incubated for 18 hrs at 37 °C. The efficiency of the digestion was assessed by running SDS-PAGE.

The peptide samples were then desalted using SepPak C18 Cartridges. The C18 cartridges were initially activated by passing through 100% Acetonitrile (ACN) and conditioned with 0.1% FA. Later, peptide mixture samples were loaded and passed through the The peptide elutions were used for further processing.

After peptide estimation, 400 μg of the peptides were taken for TMT labeling as per manufacturer's instructions (Thermo Fisher Scientific). Peptides of BEL-A BTM, BEL-A SCM, HSPCs BTM, and HSPCs SCM were labeled with 127 N, 128 N, 130 N, and 131, respectively. The labeled peptides were fractionated using the basic Reverse Phase Liquid Chromatography (bRPLC) fractionation method using C18 stage tips. A total of 24 fractions were collected and further concatenated to 6 fractions for further LC-MS/MS analysis.

2 μg peptide samples were subjected to mass spectrometer analysis in data-dependent acquisition (DDA) mode using EASY nLC 1200 nano liquid chromatography coupled to Orbitrap Fusion Tribrid (Thermo Scientific, Bremen, Germany) mass spectrometer. Each sample was resuspended with 0.1% FA and loaded onto a 96-well plate. The samples were then loaded onto the Acclaim Pep-Map™ 100 trap column (75 μm X 2 cm, nanoViper, C18, 3 μm, 100 Å) at a flow rate of 300 nL/min. The peptides were made to pass through and separated using PepMap™RSLC C18 (2 μm, 100 Å, 50 μm × 15 cm) analytical column. The column equilibration prior to each run and sample loading was performed by passing mobile phase A (0.1% FA). Peptides bound to the C18 were made to separate based on their hydrophobicity and eluted by passing mobile phase A (0.1% FA in Water) and B (0.1% FA in 80% ACN) in gradient mode for 120 min at 300 nL/min flow rate. The %B (80% acetonitrile in 0.1% formic acid) was increased gradually from 5% at 0 min to 100% at 120 min.The column temperature was set to 45 °C throughout the run.

Peptides eluted from the column were ionized in positive ion mode at EASY-Spray™ Source with Spray Voltage: 2.1 kV and Ion Transfer Tube Temp, 275 °C. Ionized peptides were acquired in data-dependent acquisition (DDA) mode with Cycle Time 5 sec. The full MS scan ranged between 400–1600 m/z precursors were acquired in Orbitrap with a resolution of 120 K. The automatic gain control (AGC) and maximum injection time (Max. IT) were set to 2e5 and 20 ms. The precursors were fragmented in the higher energy collision-induced dissociation (HCD) technique with normalized collision energy (NCE) of 35 ± 3%. Finally, the fragment ions were acquired in Orbitrap at 60 K resolution at 200 m/z.

The raw files were searched against the database Homo sapiens v110 (downloaded from NCBI) and the contaminant database in Proteome Discoverer (PD) (version 2.2, Thermo Scientific) software using SEQUEST HT and Mascot search engine. The protease enzyme Trypsin was selected with missed cleavage 1. Carbamidomethylation at Cysteine, TMT-6plex at peptide N-terminus and Lysine were set as fixed modifications. Oxidation at Methionine and Acetylation (N-terminus) were set as the dynamic modifications. Precursor and fragment mass tolerance were set as 10 ppm and 0.02 Da, respectively. The PSMs, peptides and proteins identified with a false discovery rate (FDR) of <5% (*q*-value < 0.05) were considered true positives. The PD result was quantile normalized

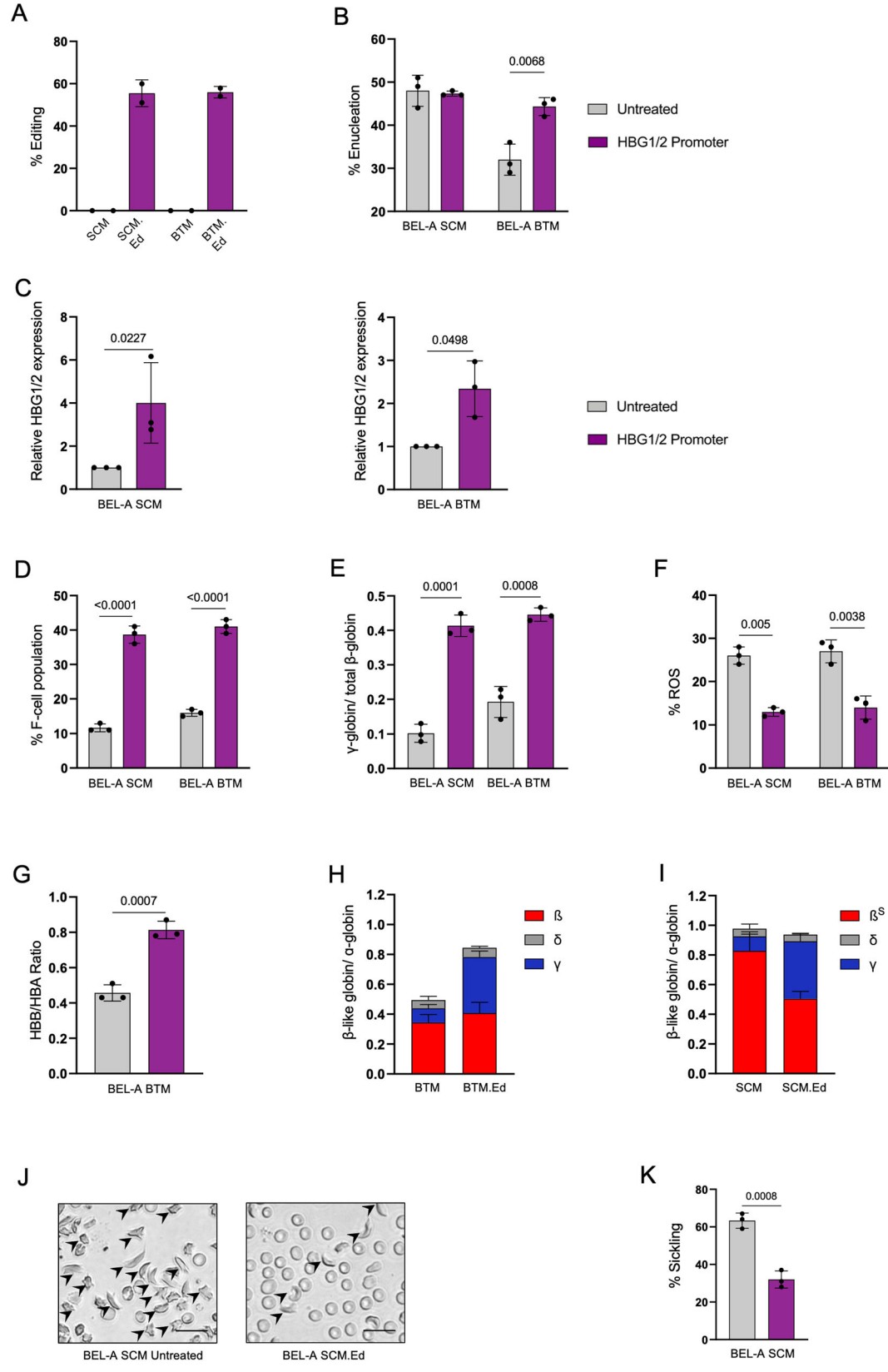

and the proteins with fold-change ratios of ≥1.5 and *p*-value ≤ 0.05 were considered to be significantly upregulated, while those with fold-change ratios of ≤0.66 and *p*-value ≤ 0.05 were considered to be significantly downregulated.

**Statistical analysis**

Statistical analysis of data was performed with GraphPad Prism Software (Version 9.5.0 GraphPad Software Inc, USA) using a two tailed student's *t*-test. Experiments were performed in triplicates

**Fig. 7 | Genome editing in *HBG1/2* promoter region rescues disease phenotype.** Genome editing was performed for disrupting the −114 to −118 position of *HBG1/2* gene (repressor binding site). **A** Indel efficiency of *HBG1/2* promoter editing in BEL-A SCM (SCM Ed.) and BEL-A BTM (BTM Ed.) cells estimated using Amplicon sequencing. **B** Enucleation percentage of unedited and *HBG1/2* promoter edited BEL-A SCM and BEL-A BTM cells (**C-D**) Estimation of increase in γ-globin gene expression. **C** Relative *HBG1/2* gene expression of unedited and *HBG1/2* promoter edited BEL-A SCM and BEL-A BTM cells. **D** Flow cytometry-based quantification of F-cell population of unedited and *HBG1/2* promoter edited BEL-A SCM and BEL-A BTM cells. **E** RP-HPLC of globin chains. Data is presented as the abundance of γ globin by the abundance of total β-like goblins (β + γ + δ). **F**–**K** Determination of rescue in disease phenotype and/or physiology at day 10 of differentiation.

**F** Percentage of ROS in unedited and *HBG1/2* promoter edited BEL-A SCM and BEL-A BTM cells. **G** Flow-cytometric analysis of β-like globin divided by α-globin in unedited and *HBG1/2* promoter edited BEL-A BTM differentiated cells. **H, I** RP-HPLC plotted as β-like globins (β + γ + δ) divided by α-globin in (**H**) BEL-A BTM unedited and *HBG1/2* promoter edited differentiated cells. **I** BEL-A SCM unedited and *HBG1/2* promoter edited differentiated cells. **J** Representative microscopic images of sickling in BEL-A SCM unedited and edited differentiated cells at Day 12 (≥200 cell counts), Scale bar: 20 μm and its (**K**) quantification. All experiments were done in triplicates independently ($n = 3$) and data is presented as Mean ± S.D. Statistical significance was determined by using two tailed student's *t*-test. Source data is provided in the Source file.

and standard errors of the mean were represented as error bars in all figures. The *P*-values at $p < 0.05$ were considered statistically significant.

### Reporting summary
Further information on research design is available in the Nature Portfolio Reporting Summary linked to this article.

## Data availability
The proteomic data generated in this study are deposited in the PRIDE database under accession code PXD044642 and processed data uploaded in Supplementary Data 1, 2. Amplicon sequencing data generated in this study are deposited in NCBI Sequencing Reads Archive (SRA) BioProject under the accession code PRJNA1070375. Raw data for each graph presented in each figure is provided in the source data file. Source data are provided with this paper.

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

## Acknowledgements

This work was supported by the Department of Science and Technology (ECR/2017/002212) Government of India (GoI), Department of Biotechnology (BT/RLF/RE-ENTRY/452013) and Council for Scientific and Industrial Research (HCP008) to S.R. S.S. was supported by NER/84/2022-ECD-I and IC-12044(11)/10/2021-ICD-DBT grants. Pr.G. and G.K. are recipients of Senior Research Fellowship from Council of Scientific and Industrial Research, and S.G.G. is recipients of Senior Research Fellowship from University Grant Commission. BEL-A cell line was created by Prof. Jan Frayne, Prof. David Anstee and Dr. Kangtana Trakarsanga with funding from the Wellcome Trust (grant numbers 087430/z/08 and 102610), NHS Blood and Transplant Department of Health (England). We thank Prof. Srinivasan Chandrasegaran from Johns Hopkins University, USA, Dr. Tanveer Ahmed from Jamia Millia Islamia, Delhi and Dr. Mohan Kumar from Centre for Stem Cell Research for critical reading of the manuscript and helpful suggestions. We also would like to thank Dr. Deepak Rathore and Translational Health Science and Technology Institute (THSTI) for providing the FACS facility and their technical expertise.

## Author contributions

S.R. conceived the study. S.R. and S.S. supervised the project. Experiments were designed by S.R., S.S., Pr.G., S.G.G. and G.K. The majority of the experiments were conducted, interpreted and analyzed by Pr.G., S.G.G. Molecular Biology and malarial invasion experiments were performed by G.K. Molecular confirmation and differentiation was performed by V.S.K. and N.B. All HPLC runs were performed by P.S., A.B.R., V.K.N., S.C.D., and T.S.K. contributed to proteomics data and interpretation. V.T.N. contributed to flow cytometric data analysis. R.C.B., A.V.R. and V.S. contributed to Amplicon sequencing and data

interpretation. S.J. and P.G. provided the samples and genotyping data. S.R., Pr.G., and S.G.G. wrote the paper with the contributions from S.S., G.K. and N.B.

## Competing interests

The authors declare no competing interest
