## [Peer Review File · Nature Communications]

REVIEWER COMMENTS

Reviewer #1 (Remarks to the Author):

A CRISPR/Cas9 coupled with piggyBac-based footprint-free approach was used to generate immortalized erythroid progenitor cells, named BEL-A SCM with a sickle hemoglobin mutation and BEL-A BTM with the IVS1-5 G>C β -thalassemia mutation. Differentiated erythroid cells from these cell lines exhibited similar differentiation profile, globin expression and proteome dynamics as primary cells. A deletion HPFH mutation was inserted in both disease lines resulting in levels of HbF that were sufficient to reverse the phenotypes of BT and SCD cells. Similar reversal of the cell phenotype was observed following disruption of the BCL11A binding site at -115 bp upstream of the HBG promoters.

These are very thorough studies convincingly supporting the claim that the BEL-A SCD and BTM cell lines will be a useful reagent for drug screening and perhaps for studying malaria.

Much of the discussion can be shortened as the effects of HbF in SCD and BTM are well known and cell-based therapeutics have provided near “cures” of these disorders by several different approaches that all produce to high levels of HbF or a HbF-like HbA. Regulatory agencies have already approved some of these treatments and others are likely to follow shortly.

Recapitulating deletion HPFH as treatment for SCD and BTM might be a fraught approach given the possibilities for rearrangements causing severe globin chain imbalance. Disruption of a transcription factor binding site in HBG promoters is already in clinical trials (NCT#04853576).

Luspatercept is approved for transfusion-dependent β thalassemia.

No need to repeat the introductory statement at the start of the Results section.

iPSCs produce HbE, not HbE, which is a variant hemoglobin.

The clinical use of hydroxyurea is not restricted because of an adverse safety profile.

Reviewer #2 (Remarks to the Author):

This paper established cell models of sickle cell disease and β -thalassemia, and used the model to verify for the first time the feasibility of the strategy of deleting large fragments of HPFH3 in the treatment of β -hemoglobinopathies.

Overall, the study presented in the manuscript is well-designed and executed, and the authors provide convincing data characterizing the newly established disease models from various aspects including cellular morphology, phenotype, molecular features, and proteomics. Moreover, the authors present a novel approach to evaluate a gene-editing therapy using the cell models. However, I have some minor concerns that the authors need to address before this manuscript can be considered for publication.

1. Is the upper section of Figure 2C depicting the SCD model and the lower section illustrating the BT model? It seems that the corresponding sample names were omitted by the author.

2. How does the author address the measurement error in the first graph of Figure 2C for the relative expression of HBB, which appears to be higher than that of other samples?
3. The discussion section has redundant information that overlaps with the introduction and should be consolidated.
4. The Method section for flow cytometry analysis contains multiple spelling errors, particularly in the last two paragraphs. Additionally, a spelling mistake is present in the final sentence of the statistical analysis.
5. Figure 4.5.7's flowchart is duplicated in Figure 1.

Reviewer #3 (Remarks to the Author):

Authors performed the proteomic analysis using high performance methods and instrumentation, however the step of protein extraction from cells is not described and the outcome of the proteomic analysis is poorly documented.

A critical point in the described analysis is its first step the cell lysis. Authors refer to Reference 70 Misra et al. describing analysis of serum samples. Whereas serum contains well soluble proteins cells contain a significant portion of membrane proteins that require harsh denaturing condition for extraction by use of detergents. This raises the question whether the extraction was complete. Most probably not. A consequence of a partial extraction is an incomplete proteome and quantitative differences in protein abundances between different cells may be biased.

Considering that the proteomic analysis involved ion-exchange fractionation of peptides prior reverse phase separation and MS analysis the number of 1000 identified protein is low. For comparison, Gautier et al identified 7,361 proteins expressed in primary erythroid cells and quantified 6,130 of these at copy numbers per cell [Cell Reports 16 (5), 1470-1484]. Authors have to present the complete proteomic data and calculate protein abundances and compare these with data present in literature.

Point-by-point response to reviewers:

Development of pathophysiologically relevant models of sickle cell disease and β -thalassemia for therapeutic studies (NCOMMS-23-06493)

We express our gratitude to all the reviewers for reviewing our manuscript and providing us with their valuable comments. We were glad to receive positive feedback from the reviewers regarding our work. In this document, we provide a point-by-point response to the comments and the new information that we have included to improve the quality of our manuscript. **The revised manuscript provides a comprehensive account of the additions and corrections, highlighted in yellow.** We are grateful for the reviewers' efforts in assisting us with improving our manuscript.

Reviewer #1:

These are very thorough studies convincingly supporting the claim that the BEL-A SCD and BTM cell lines will be a useful reagent for drug screening and perhaps for studying malaria.

We thank the reviewer for the positive comment on the manuscript.

Much of the discussion can be shortened as the effects of HbF in SCD and BTM are well known and cell-based therapeutics have provided near "cures" of these disorders by several different approaches that all produce to high levels of HbF or a HbF-like HbA.

As suggested by the reviewer, we have shortened the discussion related to HbF in the manuscript and updated the discussion to be compendious.

Luspatercept is approved for transfusion-dependent β -thalassemia.

We apologize for missing to include Luspatercept. We have added the same to the manuscript.

No need to repeat the introductory statement at the start of the Results section.

As suggested by the reviewer, we have removed the introductory statement at the start of the results section.

iPSCs produce HBE, not HbE, which is a variant hemoglobin.

We apologize for the mistake in nomenclature. We have corrected this in the manuscript

The clinical use of hydroxyurea is not restricted because of an adverse safety profile.

We thank the reviewer for this suggestion. We corrected this sentence in the manuscript

Studies have raised concerns regarding the safety profile of HU due to its associations with thrombocytopenia, myelotoxicity, and higher infection rates (44,45).

Reviewer #2:

This paper established cell models of sickle cell disease and β -thalassemia, and used the model to verify for the first time the feasibility of the strategy of deleting large fragments of HPFH3 in the treatment of β -hemoglobinopathies.

Overall, the study presented in the manuscript is well-designed and executed, and the authors provide convincing data characterizing the newly established disease models from various aspects including cellular morphology, phenotype, molecular features, and proteomics. Moreover, the authors present a novel approach to evaluate a gene-editing therapy using the cell models. However, I have some minor concerns that the authors need to address before this manuscript can be considered for publication.

We thank the reviewer for his/her very positive feedback on the manuscript and for providing very helpful comments.

1. Is the upper section of Figure 2C depicting the SCD model and the lower section illustrating the BT model? It seems that the corresponding sample names were omitted by the author.

We apologize that this discrepancy wasn't explained clearly in the manuscript. We have now clarified this error by updating the detail of each bar in Figure 2C.

2. How does the author address the measurement error in the first graph of Figure 2C for the relative expression of HBB, which appears to be higher than that of other samples?

We thank the reviewer for pointing out this.

Prompted by the reviewer's comment, we repeated this experiment and measured the relative expression of the beta-globin gene in BEL-A SCM and primary SCM HSPCs. We observed the expression of the beta-globin gene was reduced in both BEL-A SCM and SCM HSPCs. We have updated the same in the manuscript Results Section.

3. The discussion section has redundant information that overlaps with the introduction and should be consolidated.

We removed the redundant information in the discussion that overlaps with the introduction and consolidated the same.

4. The Method section for flow cytometry analysis contains multiple spelling errors, particularly in the last two paragraphs. Additionally, a spelling mistake is present in the final sentence of the statistical analysis.

We thank the reviewer for this suggestion, we meticulously went through the entire manuscript carefully and have corrected all the grammar and spelling errors.

5. Figure 4.5.7's flowchart is duplicated in Figure 1.

As suggested by the reviewer, we have removed the duplication of the flowchart from Figure 4, 5 and 7.

Reviewer #3:

Authors performed the proteomic analysis using high performance methods and instrumentation, however the step of protein extraction from cells is not described and the outcome of the proteomic analysis is poorly documented. A critical point in the described analysis is its first step, the cell lysis. Authors refer to Reference 70 Misra et al. describing analysis of serum samples. Whereas serum contains well soluble proteins cells contain a significant portion of membrane proteins that require harsh denaturing conditions for extraction by use of detergents. This raises the question whether the extraction was complete. Most probably not. A consequence of a partial extraction is an incomplete proteome and quantitative differences in protein abundances between different cells may be biased. Considering that the proteomic analysis involved ion-exchange fractionation

of peptides prior reverse phase separation and MS analysis the number of 1000 identified proteins is low. For comparison, Gautier et al identified 7,361 proteins expressed in primary erythroid cells and quantified 6,130 of these at copy numbers per cell [Cell Reports 16 (5), 1470-1484]. Authors have to present the complete proteomic data and calculate protein abundances and compare these with data present in the literature.

We thank the reviewer for providing us with very helpful comments and suggestions.

We apologize for the lack of information. We have repeated the proteomics experiment with a different protein extraction method and TMT labeling. In the revised manuscript, we provided a more detailed experimental methodology of sample preparation and TMT LC-MS/MS methodology.

All the concerns raised by the reviewer 3 have been addressed below.

Protein extraction and estimation:

The cell pellets were resuspended in 2% lysis buffer (1X Halt Protease and Phosphatase inhibitor cocktail, 2% SDS and 50mM TEAB) in cold conditions. The samples were then sonicated, centrifuged and supernatants were collected. The protein content in the samples was measured using the Pierce™ BCA Protein Assay kit as per the manufacturer's protocol. Following the BCA estimation, an SDS-PAGE analysis was conducted for quality control. Subsequently, 400µg of protein was subjected to reduction and alkylation using 10mM DTT and 20mM IAA respectively. The protein was precipitated by adding 6X volume of chilled acetone and kept for overnight incubation at -20°C. The protein samples were digested using Trypsin (Promega) in the ratio 1:20 and incubated for 18 hrs at 37°C. The efficiency of the digestion was assessed by running SDS-PAGE.

Desalting and peptide estimation:

The peptide samples were then desalted using SepPak C18 Cartridges. The C18 cartridges were initially activated by passing through 100% Acetonitrile (ACN) and conditioned with 0.1% FA. Later, peptide mixture samples were loaded and passed through the The peptide elutions were used for further processing.

The dried eluates were were then quantified using Peirce Quantitative Colorimetric Peptide Assay (Thermo Scientific) as per manufacturer's instructions.

Tandem Mass Tag (TMT) Labelling and fractionation:

After peptide estimation, 400ug of the peptides were labelled with TMT labelling as per manufacturer's protocol (Thermo Fisher Scientific). The labelled peptides were fractionated using basic Reverse Phase Liquid Chromatography (bRPLC) fractionation

method. The fractionation was carried out using C18 stage tips. 24 fractions were concatenated to 6 fractions.

The TMT labels used are as followed:

Sl. No.	Name of the samples	TMT labels used
1	BELA - BTM	127N
2	BELA - SCM	128N
3	HSPC - BTM	130N
4	HSPC - SCM	131

LC-MS/MS Analysis:

Based on the peptide estimation around 2ug of the peptide were subjected to mass spectrometer analysis in data-dependent acquisition (DDA) mode using EASY nLC 1200 nano liquid chromatography coupled to Orbitrap Fusion Tribrid mass spectrometer (Thermo Scientific, Bremen, Germany). Each sample was resuspended with 0.1% FA and loaded onto 96-well plate. The samples were loaded onto the Acclaim PepMap™ 100 trap column (75µm X 2cm, nanoViper, C18, 3µm, 100Å) at a flow rate of 300 nL/min. The peptides were made to pass through and separated using PepMap™ RSLC C18 (2µm, 100Å, 50 µm × 15 cm) analytical column. The column equilibration prior to each run and sample loading was performed by passing mobile phase A (0.1% FA). Peptides bound to the C18 were made to separate based on their hydrophobicity and eluted by passing mobile phase A (0.1% FA in Water) and B (0.1% FA in 80% ACN) in gradient mode for 120 min at 300 nL/min flow rate. The %B was increased gradually from 5% at 0 min to 100% in 120 min. The column temperature was set to 45°C throughout the run.

Peptides eluted from the column were ionized in positive ion mode at EASY-Spray™ Source with Spray Voltage: 2.1 kV and Ion Transfer Tube Temp, 275°C. Ionized peptides were acquired in data-dependent acquisition (DDA) mode with Cycle Time 5 sec. Under Full MS scan, peptide precursors ranging between 400– 1600 m/z were acquired in Orbitrap with the resolution of 120 K. The automatic gain control (AGC) and maximum injection time (Max. IT) were set to 2e5 and 20 ms. Precursors with charge state (z) between 2-7 was separated in Quadrupole. The precursors were fragmented in the higher energy collision-induced dissociation (HCD) technique with normalized collision energy (NCE) of 35±3%. Finally, the fragment ions were acquired in Orbitrap at 30 K resolution with 'Auto Normal' scan range mode. AGC and Max. IT were set to 1e5 and 200 ms, respectively.

Data analysis:

The raw files were processed for data analysis with Proteome Discoverer (PD) software v2.2 (Thermo Scientific) and a search was performed against *Homo sapiens* v110 database (downloaded from NCBI) using SEQUEST HT and Mascot search engines. The protease enzyme Trypsin was selected with missed cleavage 1. Carbamidomethylation at Cysteine, TMT-6plex at peptide N-terminus and Lysine were set as fixed modifications. Oxidation at Methionine and Acetylation (N-terminus) were set as the dynamic modifications. Precursor and fragment mass tolerance were set as 10 ppm and 0.02 Da, respectively. The PSMs, peptides and proteins identified with a false discovery rate (FDR) of less than 5% (q-value < 0.05) were considered true positives. The PD result was quantile normalized and the proteins with fold-change ratios of ≥ 1.5 and p-value ≤ 0.05 were considered to be significantly upregulated, while those with fold-change ratios of ≤ 0.66 and p-value ≤ 0.05 were considered to be significantly downregulated.

In the present study, we analyzed the proteome at the proerythroblast stage and obtained 4943 proteins. Gautier et. al. ([Cell Reports 16 (5), 1470-1484) have also shown around 5000 proteins at proerythroblast stage.

As suggested by the reviewer, the proteomic data obtained through mass spectrometry have been submitted to the ProteomeXchange consortium using the PRIDE (Perez-Rivertol et al 2022) partner repository. We have also provided Proteomics Data in Supplementary Table 1 and 2

We thank him/her for the constructive comments. We have experimentally addressed all of her/his concerns.

REVIEWERS' COMMENTS

Reviewer #1 (Remarks to the Author):

The authors have been responsive to my comments. My sole suggestion is that they amend the end of the first paragraph of the introduction to acknowledge that one form of gene therapy is approved for thalassemia (another is very likely to soon be approved) and that two types of gene therapy are now widely approved for treating SCD.

Reviewer #2 (Remarks to the Author):

After a thorough review of the revised manuscript and careful consideration of the responses provided by the authors to the queries raised during the initial review, I am pleased to report that all of the previously outlined concerns have been satisfactorily addressed.

Reviewer #3 (Remarks to the Author):

Authors have improved the proteomic analysis. They described in detail how it was conducted and provided complete proteomic data. Now the depth of analysis is sufficient to make conclusions presented in the manuscript.

Point-by-point response to reviewers:

Development of pathophysiologically relevant models of sickle cell disease and β -thalassemia for therapeutic studies (NCOMMS-23-06493A)

We express our gratitude to all the reviewers for reviewing our manuscript and providing us with their valuable comments. We were glad to receive positive feedback from all the reviewers regarding our work. We are grateful for the reviewers' efforts in assisting us with significantly improving our manuscript.

Reviewer #1:

The authors have been responsive to my comments. My sole suggestion is that they amend the end of the first paragraph of the introduction to acknowledge that one form of gene therapy is approved for thalassemia (another is very likely to soon be approved) and that two types of gene therapy are now widely approved for treating SCD.

We thank the reviewer for this suggestion. It has been amended at the end of the first paragraph of the introduction.

Reviewer #2:

After a thorough review of the revised manuscript and careful consideration of the responses provided by the authors to the queries raised during the initial review, I am pleased to report that all of the previously outlined concerns have been satisfactorily addressed.

We are grateful to Reviewer #2 for providing positive feedback on our revised manuscript and appreciate his/her dedicated effort in assessing our work.

Reviewer: #3

Authors have improved the proteomic analysis. They described in detail how it was conducted and provided complete proteomic data. Now the depth of analysis is sufficient to make conclusions presented in the manuscript.

We are delighted to learn that our revised version of the manuscript satisfied Reviewer #3 concerns and thank Reviewer # 3 for his/her effort in evaluating our work.